# LIBRA: EFFECTIVE YET EFFICIENT LOAD BALANCING FOR LARGE-SCALE MoE INFERENCE

**Jaehoon Yang, Yushin Kim, Seokwon Moon, Yeonhong Park, Jae W. Lee**
Seoul National University
{jaehoon.yang,henry11n,moonsw214,ilil96,jaewlee}@snu.ac.kr

## ABSTRACT

Distributed inference of large-scale Mixture-of-Experts (MoE) models faces a critical challenge: expert load imbalance. Numerous system-level approaches have been proposed for load balancing, but they either fail to achieve a satisfactory level of balance or introduce new bottlenecks due to the overhead of the load balancing mechanism itself. To this end, we propose *Libra*, a system that achieves near-optimal load balancing with minimal overhead. *Libra* adopts sophisticated mechanisms that accurately predict future expert activations and, based on these predictions, systematically perform load balancing. At the same time, it effectively hides the associated overhead by reconstructing the execution flow so that these costs are overlapped with MoE computation. Evaluations with two large-scale state-of-the-art MoE models on 8 H200 GPUs demonstrate that *Libra* improves throughput by up to 19.2%. The code is available at https://github.com/SNU-ARC/Libra.

## 1 INTRODUCTION

The Mixture-of-Experts (MoE) architecture has become a cornerstone for state-of-the-art Large Language Models (LLMs) such as DeepSeek-V3, Qwen3MoE, and GLM-4.5 (DeepSeek-AI et al., 2025; Yang et al., 2025; GLM-4.5 Team et al., 2025). Through sparse activation, MoE enables models to scale to trillions of parameters while keeping the training and inference computation cost manageable (Du et al., 2022; The Mosaic Research Team, 2024; Jiang et al., 2024; Fedus et al., 2022; Lepikhin et al., 2020; Rajbhandari et al., 2022).

At the same time, the dynamic nature of MoE models introduces a key deployment challenge: *load imbalance*. One common way to scale MoE inference is through Expert Parallelism (EP), in which experts within MoE layers are partitioned across multiple GPUs. Under this setup, load imbalance arises when a disproportionate number of tokens are assigned to a few *hot* experts, causing the GPUs hosting them to become stragglers that determine the end-to-end latency.

Existing system-level load balancing approaches suffer from fundamental limitations, proving to be less effective and/or inefficient (DeepSeek, 2025; Li et al., 2023; Doucet et al., 2025). Some approaches fail to achieve satisfactory balance because they rely on ineffective heuristics, leaving substantial room for improvement (DeepSeek, 2025; Li et al., 2023). Others achieve a considerable level of balance but introduce new bottlenecks due to the additional operations required for load balancing (Doucet et al., 2025).

To address these challenges, we propose *Libra*, a system that achieves near-optimal balance with virtually zero overhead. In other words, it catches two birds at once: effective load balancing and efficient realization of that mechanism. For effectiveness, Libra predicts expert activations for the next layer with high accuracy by leveraging the observation that hidden states in LLMs evolve slowly across consecutive blocks, and based on these predictions, applies a sophisticated algorithm that yields near-optimal balance. For efficiency, Libra reconstructs the inference execution flow so that any overhead incurred by this process is hidden under MoE computations. In evaluations on eight benchmarks using two state-of-the-art MoE models, Qwen3MoE and GLM-4.5, on 8 H200 GPUs, Libra improves throughput by up to 19.2% compared to the state of the art.

## 2 BACKGROUND AND MOTIVATION

### 2.1 EXPERT LOAD IMBALANCE IN MoE INFERENCE

The Mixture-of-Experts (MoE) (Jacobs et al., 1991; Jordan & Jacobs, 1994; Shazeer et al., 2017) architecture enhances the capacity of Large Language Models (LLMs) by replacing the dense Feed-Forward Network (FFN) layer in a Transformer block with a sparse MoE layer. This layer consists of a large pool of subnetworks (experts) and a gating network that selectively activates a small subset of experts (e.g., top-k) for each input token. This sparse activation allows MoE models to scale to hundreds of billions or even trillions of parameters while keeping the computational cost for inference relatively low (Du et al., 2022; The Mosaic Research Team, 2024; Jiang et al., 2024; Fedus et al., 2022; Rajbhandari et al., 2022; Lepikhin et al., 2020). Consequently, large-scale open-source MoE models have achieved performance comparable to leading proprietary models like GPT-5, demonstrating the efficacy of this architecture (Yang et al., 2025; DeepSeek-AI et al., 2025; GLM-4.5 Team et al., 2025; Baidu ERNIE Team, 2025; OpenAI et al., 2025; Kimi Team et al., 2025).

However, the inherent mechanism that grants MoE models their efficiency—independent token assignment—introduces a significant challenge: *expert load imbalance*. Historically, this issue was addressed during the training phase by incorporating an auxiliary load-balancing loss term, which encouraged a more uniform distribution of tokens across all experts (Xue et al., 2024; Muennighoff et al., 2025; Fedus et al., 2022). While effective for balancing, this approach often came at the cost of model performance, as it could hinder the degree of expert specialization (Wang et al., 2024; Guo et al., 2025; Qiu et al., 2025; DeepSeek-AI et al., 2025).

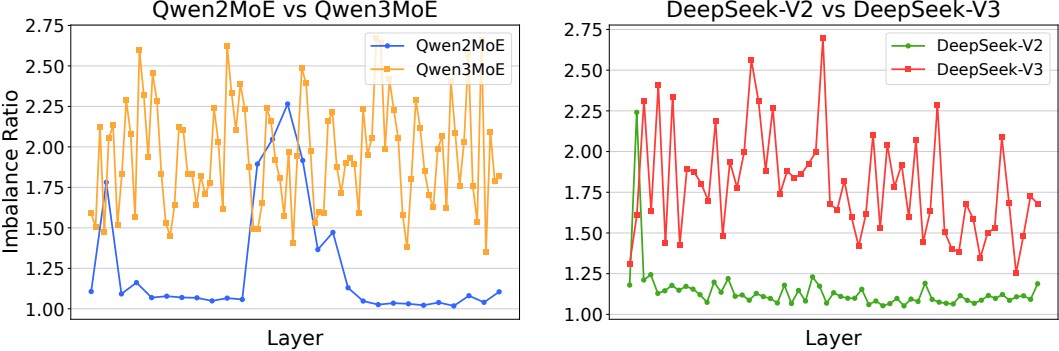

Figure 1: Intensified expert load imbalance in recent MoE models.

Reflecting this trade-off, recent state-of-the-art MoE models have moved away from strict load-balancing loss in favor of techniques that maximize expert specialization (Yang et al., 2025; GLM-4.5 Team et al., 2025; DeepSeek-AI et al., 2025). This aggressive pursuit of specialization has successfully pushed model performance to new heights but has the critical side effect of intensifying the expert load imbalance during inference. We measure this using the imbalance ratio, defined as the maximum load on any single GPU divided by the average load across all GPUs, where a value of 1.0 indicates a perfect balance. This trend is illustrated in Figure 1, showing a stark contrast in the imbalance ratio between newer MoE models and their predecessors (see Appendix A for experimental setup details). This reveals a fundamental trade-off: *achieving state-of-the-art performance in large MoE models exacerbates the expert load imbalance*, a problem projected to become more severe as models advance.

To efficiently serve large-scale MoE models across multiple GPUs, the standard strategy is a hybrid approach that applies Expert Parallelism (EP) to MoE layers and Data Parallelism (DP) to non-MoE layers (e.g., self-attention) (Perplexity AI, 2025; SGLang Team, 2025; Doucet et al., 2025; Li et al., 2025). While this strategy is essential for managing the massive parameter counts of these models, its performance is highly susceptible to expert load imbalance, which consequently becomes a critical performance bottleneck (Doucet et al., 2025). This issue stems from the synchronous execution of the MoE layers, forcing all devices to wait for the most heavily loaded GPU—the one hosting the hot expert(s)—to finish its computation. This phenomenon, known as the *straggler effect*, leads

to significant idle time on less-loaded workers, severely degrading end-to-end latency and overall system throughput. Consequently, mitigating this straggler effect has become a central challenge in MoE inference, prompting the exploration of various system-level solutions.

## 2.2 SYSTEM-LEVEL SOLUTIONS FOR LOAD BALANCING AND LIMITATIONS

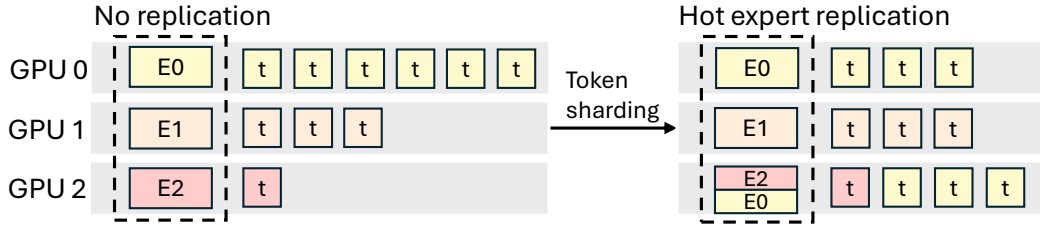

Figure 2: Hot expert replication and token sharding

A number of studies have attempted to mitigate load imbalance in MoE inference through system-level techniques. Most of these approaches employ *hot expert replication*. Figure 2 illustrates this approach: instead of assigning each expert to a unique GPU without redundancy, hot experts (i.e., experts that are likely to receive many tokens) are replicated across multiple GPUs. During MoE execution, tokens routed to these hot experts can then be distributed across replicas on different GPUs—a process we refer to as *token sharding*. This alleviates bottlenecks that would otherwise arise if a single GPU were forced to process a disproportionate number of tokens.

In the following, we discuss three representative works that constitute the most widely used and/or state-of-the-art techniques. They share the above approach but differ in how they perform expert replication and token sharding. While these methods present promising results, they also exhibit notable limitations, as their strategies for replication and sharding achieve only limited effectiveness and/or efficiency.

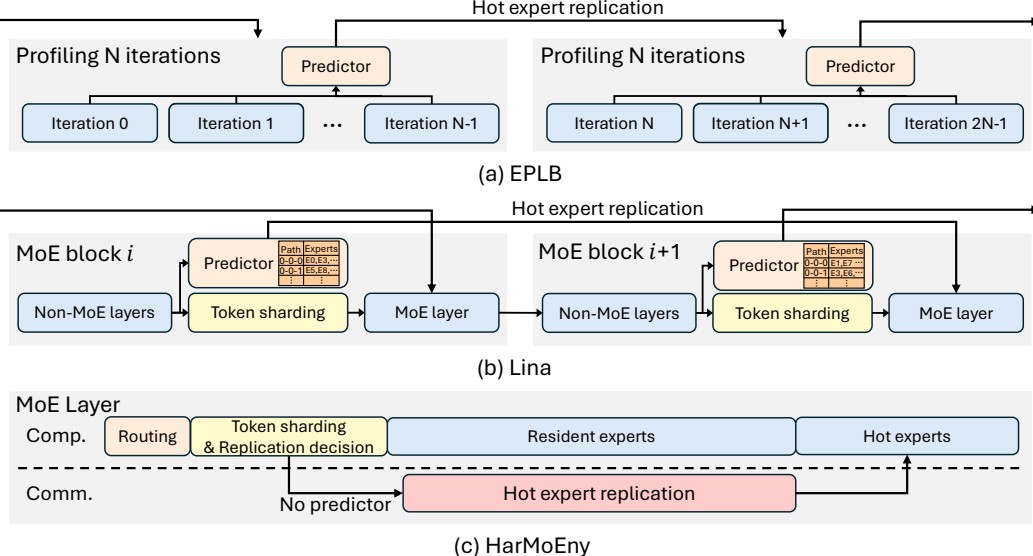

Figure 3: Overview of EPLB, Lina, and HarMoEny

**EPLB**. Expert Placement Load Balancer (EPLB) (DeepSeek, 2025) periodically performs expert replication based on historical data. Figure 3(a) illustrates this process. It profiles expert popularity over a fixed number of iterations (N in the figure) and uses the aggregated statistics to guide expert replication. However, this strategy is less effective because past popularity does not capture the instantaneous and dynamic variations across requests. EPLB's token sharding is also limited in

effectiveness: once experts are replicated, tokens routed to them are randomly distributed across the GPUs holding replicas. In short, while EPLB is relatively efficient, it is not effective in either expert replication or token sharding.

**Lina**. Figure 3(b) illustrates how Lina (Li et al., 2023) performs expert replication. Specifically, Lina replicates experts for the upcoming block: when executing MoE computation for Block $i$, it performs expert replication for Block $i + 1$, thereby removing replication overhead from the critical path. For this purpose, it relies on a pre-constructed lookup table that tracks each token's expert-selection-path. This path, which is the sequence of experts a token has selected in previous few layers (e.g., from layer $i - 4$ to layer $i - 1$), is used to predict which experts will be popular in the current layer (layer $i$). Such expert replication, however, is limited in effectiveness. In our evaluations on eight different benchmarks with two models (Qwen3MoE and GLM-4.5), Lina's prediction accuracy falls to as low as 43.7% and 11.8%, respectively. Details of the experimental setup are provided in Appendix A.

For token sharding, Lina simply distributes tokens uniformly across all replicas (e.g., assigning an equal number of tokens to each). Similar to the random token sharding in EPLB, this strategy is largely oblivious to the actual GPU loads, leaving significant room for improvement.

**HarMoEny**. Figure 3(c) illustrates HarMoEny (Doucet et al., 2025), which makes decisions on expert replication based on exact routing results. In other words, expert replication is performed only after the current input and its routing outcomes become available. This design makes replication highly effective, as it is guided by exact information. To prevent replication overhead from appearing on the critical path, HarMoEny first performs MoE computation for tokens routed to resident experts, while replication of hot experts proceeds in parallel. Once replication completes, the MoE computation for hot experts is executed. Token sharding is also effective, as HarMoEny employs a sophisticated algorithm that computes a near-optimal token assignment.

Despite this effectiveness, HarMoEny suffers from efficiency issues. The overhead required to realize such accurate replication and token sharding is substantial. After routing and before completing MoE computation, HarMoEny must perform decision making through complex algorithms to determine both expert replication and token sharding. Because these algorithms run synchronously on the GPU, they extend the critical path and introduce new bottlenecks.

## 3 DESIGN

In this section, we propose Libra, a system for MoE inference that achieves near-optimal load balancing with minimal overhead. Unlike prior methods, Libra simultaneously addresses both effectiveness and efficiency in hot expert replication and token sharding, thereby overcoming the key limitations of existing approaches. For expert replication, Libra follows the spirit of Lina by prefetching hot experts for the next layer while processing the current layer, based on prediction. This design avoids the inefficiency observed in HarMoEny, which cannot exploit Grouped-GEMM optimizations. At the same time, Libra employs a more accurate prediction mechanism than Lina, thereby improving effectiveness. For token sharding, Libra adopts the strategy of HarMoEny but effectively hides its cost from the critical path by restructuring the execution flow and leveraging the CPU.

The remainder of this section is organized as follows. Section 3.1 explains Libra's execution flow. Section 3.2 describes the hot expert replication mechanism, and Section 3.3 details the token sharding mechanism.

### 3.1 LIBRA EXECUTION FLOW

Figure 4 illustrates the execution flow of Libra. The key novelty lies in its *Two-Stage Locality-Aware Execution*, which splits MoE computation into two phases based on token locality: $MoE_{local}$ and $MoE_{remote}$. The $MoE_{local}$ phase processes tokens routed to experts residing on the same GPU as the tokens themselves, while the $MoE_{remote}$ phase handles tokens that must be dispatched to other GPUs. After decomposing the computation into these two phases, Libra first performs $MoE_{local}$, followed by $MoE_{remote}$.

This execution flow creates a time window in which the overhead of sophisticated token sharding mechanism can be hidden. In the conventional execution flow with load balancing, MoE computation begins only after token sharding completes. By contrast, in Libra, the $MoE_{local}$ phase has

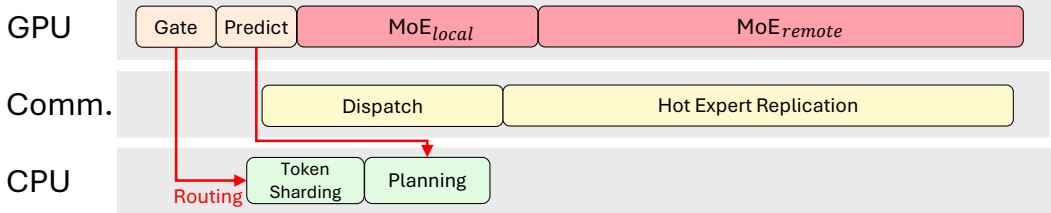

Figure 4: Two-Stage Locality-Aware Execution of Libra

no dependency on token sharding; it can start immediately after the gating function, with only the $MoE_{remote}$ phase depending on the results of token sharding and Dispatch operation. This enables token sharding mechanism to run in parallel with $MoE_{local}$.

To further enhance the effectiveness of this parallelism, Libra performs token sharding on the CPU rather than the GPU. In addition, Libra implements dispatch using AllGather (i.e., all tokens are transferred to all GPUs) instead of All2All, where tokens are sent only to their assigned GPU. While this design increases the raw communication volume of dispatch, its latency impact is negligible. More importantly, it improves efficiency by removing dispatch from the critical path: in an All2All-based implementation, dispatch must wait for token sharding to finish, whereas in the AllGather-based implementation, dispatch can also proceed in parallel with token sharding.

## 3.2 HOT EXPERT REPLICATION

As mentioned, Libra follows the spirit of Lina for hot expert replication: performing expert replication for the next layer while processing the current layer, based on prediction. However, Libra departs substantially from Lina in both how the prediction is performed and how expert replication planning (i.e., determining which experts to replicate to which GPUs) is carried out.

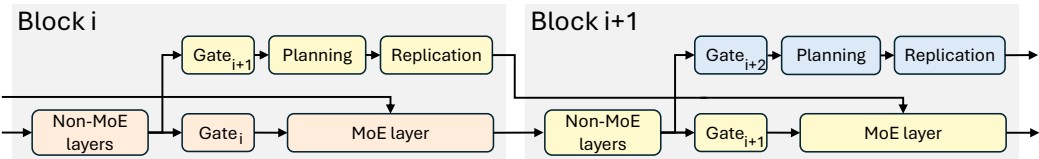

Figure 5: Hot expert replication of Libra with lookahead predictors

**Predictor Design**. Libra employs a lookahead predictor, leveraging a well-established property of Transformer-based LLMs: hidden states evolve slowly across layers (Liu et al., 2023b; Hwang et al., 2024). Figure 5 illustrates its concept. It speculatively executes the gating function of the next layer using the hidden states from the current layer, and then uses the results to determine which experts should be replicated across GPUs. This runtime-based approach achieves substantially higher accuracy than Lina's predictor (e.g. 70-80% vs 20-30%).

**Locality-Aware Expert Replication Planning**. During expert replication planning, Libra introduces an additional consideration beyond load balancing: locality enhancement. In other words, Libra not only balances load but also seeks to extend the $MoE_{local}$ computation window, thereby providing more opportunity to hide token sharding overhead.

To this end, Libra performs expert replication planning in two phases. In the first phase, each GPU brings in $N \times \alpha$ experts that are most frequently activated by the tokens on that GPU and are not already resident on it, thereby extending the $MoE_{local}$ computation window. Here, $N$ denotes the maximum number of additional experts a GPU may host, determined by its available memory capacity and the allowable time window (a function of MoE computation time and communication bandwidth), while $\alpha$ is a hyperparameter that controls what fraction of $N$ is allocated to the first phase. In the second phase, load balancing is performed iteratively: at each step, the hottest expert

from the most heavily loaded GPU is selected for replication and placed on the least-loaded GPU among those that have not yet received $N$ extra experts.

Figure 6 illustrates the expert replication planning process of Libra with an example. First, to enhance locality, Libra addresses the initial placement (left), where a significant portion of tokens on each GPU is routed to remote experts on other devices. Each GPU identifies its most requested remote experts and replicates them locally. For instance, GPU 0 replicates E4 from GPU 2 to serve its local tokens. This process converts remote tokens into local computations, securing the $MoE_{local}$ computation window necessary to hide system overhead. Second, to establish a foundation for load balancing, the algorithm identifies and replicates heavily loaded experts to under-loaded GPUs. In the figure, this is shown by replicating expert E2 to an under-loaded device. This facilitates effective token sharding by allowing the workload from overloaded GPUs to be redistributed, ultimately enabling a near-perfect load balance.

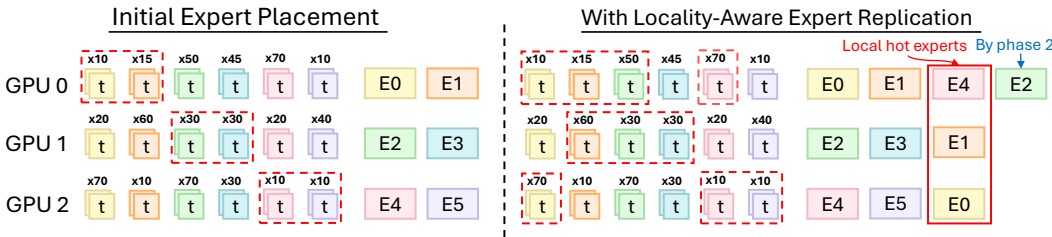

Figure 6: Locality-aware expert replication planning

## 3.3 ADAPTIVE TOKEN SHARDING

For token sharding, Libra adopts an algorithm similar to that used by HarMoEny, but with two key differences. First, Libra applies token sharding only to remote tokens. Second, this process is offloaded to the CPU.

Figure 7 explains the iterative greedy strategy of token sharding. The algorithm's main loop begins by checking if any GPU's load exceeds the target threshold (❷). If the system is balanced, the process terminates. Otherwise, it selects the most overloaded GPU, $g_s$, to resolve (❸). To find the most effective transfer, the algorithm enters an inner loop, starting by selecting the hottest remote expert, $e$, on $g_s$—the one accounting for the largest number of its remote tokens (❹). It then searches for an optimal destination: the least-loaded GPU ($g_d$) that hosts a replica of expert $e$ and has enough capacity to accept new tokens (❺). A replica of the expert is necessary on the destination GPU to process the transferred tokens, and these replicas are enabled by hot expert replication. If a suitable destination is found, the algorithm calculates the number of tokens to transfer and updates the loads on both $g_s$ and $g_d$ (❻). Crucially, after each successful transfer, the algorithm returns to the main loop's start (❷) to re-evaluate the entire system's balance, ensuring it always ad-

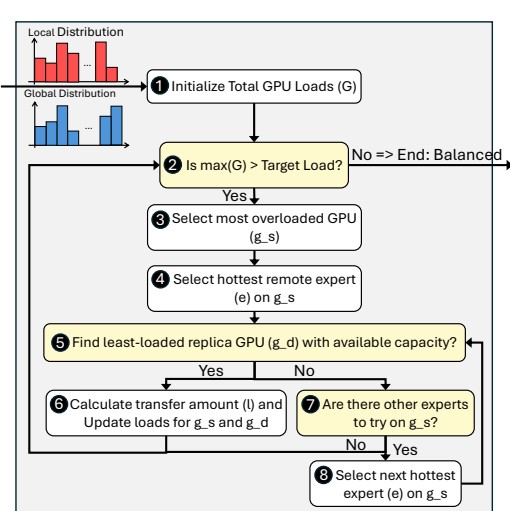

Figure 7: Logic flow of the iterative greedy rebalancing strategy.

dresses the most critical imbalance first. If no suitable destination is found (❼), it attempts to transfer the tokens for the next hottest expert on $g_s$ (❽) until all options are exhausted, at which point it also returns to re-evaluate the global state. The full algorithm is detailed in Appendix C.

## 4 IMPLEMENTATION DETAILS

We implement Libra atop the SGLang (v0.4.10) LLM serving framework. Our core mechanisms for expert replication planning and token sharding are implemented in Cython to ensure minimal overhead and are integrated as native modules during SGLang's build process.

To efficiently perform hot expert replication, we leverage PyTorch `SymmetricMemory` for copy engine-based P2P transfers. We employ a double-buffering strategy by pre-allocating two large buffers: an even buffer and an odd buffer. During the execution of an even-numbered MoE layer, original and duplicated experts are gathered in the even buffer to be processed via a high-performance Grouped-GEMM kernel. Concurrently, the system loads the necessary experts for the subsequent odd-numbered layer into the odd buffer. When processing an odd-numbered MoE layer, the roles are reversed: computation utilizes the odd buffer while the even buffer is populated for the next layer. This pipelining mechanism effectively hides the expert replication overhead by overlapping the P2P copy operations with the ongoing computation.

## 5 EVALUATION

We conduct a comprehensive evaluation to demonstrate the effectiveness of Libra. Our experiments are designed to answer four key questions: (1) How does Libra's prefill performance compare against baselines? (2) How stable and robust is Libra's performance under workloads with dynamic and shifting token distributions? (3) How does the prediction accuracy of Libra's speculative execution compare against existing methods like Lina? (4) What is the overhead introduced by Libra's mechanisms, and how effectively is it hidden within the execution flow?

### 5.1 SETUP

**Model and Data**. We evaluate Libra using two representative state-of-the-art large MoE models: Qwen3MoE (235B) (Yang et al., 2025) and GLM-4.5 (355B) (GLM-4.5 Team et al., 2025). To ensure coverage of a wide range of inputs, we use eight datasets: BookCorpus (Zhu et al., 2015), Codeforces (Penedo et al., 2025), DeepSeek-Prover (Xin et al., 2024), FineWeb (Penedo et al., 2024), GSM8K (Cobbe et al., 2021), HellaSwag (Zellers et al., 2019), HumanEvalPlus (Liu et al., 2023a), and LMSYS-Chat-1M (Zheng et al., 2023). All experiments are run using BF16 precision.

**Environments**. All experiments are conducted on a single node equipped with 8 NVIDIA H200-SXM5 GPUs, each with 141 GB of HBM3e memory. Intra-node communication leverages NVSwitch with 900 GB/s of P2P bandwidth.

**Baselines**. We compare Libra against three baselines. The vanilla MoE implementation in SGLang (v0.4.10) serves as our foundational baseline, representing a standard system without advanced load balancing. For the widely adopted proactive expert replication approach, we use EPLB (DeepSeek, 2025) from its implementation within SGLang. As the strongest baseline, we evaluate against Lina (Li et al., 2023). Since no public implementation of Lina is available, we developed an in-house version built on SGLang, faithfully following the description in the original paper.

**Metrics**. The primary performance metric is prefill throughput, measured in tokens per second. We assume a prefill-decode disaggregated serving system where the prefill and decode phases are separated and handled by different GPUs (Zhong et al., 2024b; Hu et al., 2025; Feng et al., 2025), and therefore target only the prefill phase in our evaluation. We also measure the imbalance ratio (defined as the load of the most burdened GPU divided by the average load across all GPUs) to analyze the effectiveness of load balancing.

### 5.2 RESULTS

**Throughput Results**. First, Libra substantially improves the performance of the prefill phase. As shown in Figure 8, Libra achieves the highest throughput across all tested models and datasets. To rigorously evaluate the robustness of our system, we focus on the four datasets exhibiting the most severe expert load imbalance. These challenging scenarios serve as stress tests to clearly distinguish the efficacy of different load balancing strategies. Notably, this evaluation was conducted under an experimental setup deliberately designed to be highly advantageous for the baseline systems. For

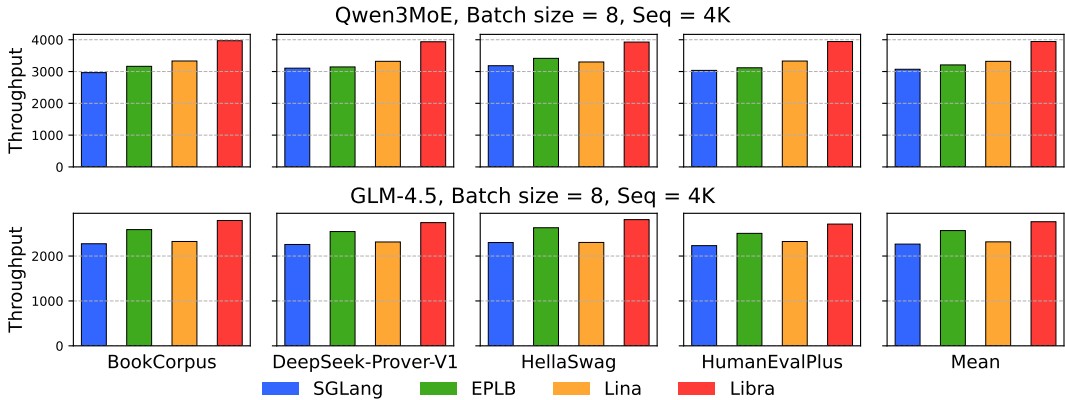

Figure 8: Prefill throughput of Libra and baselines.

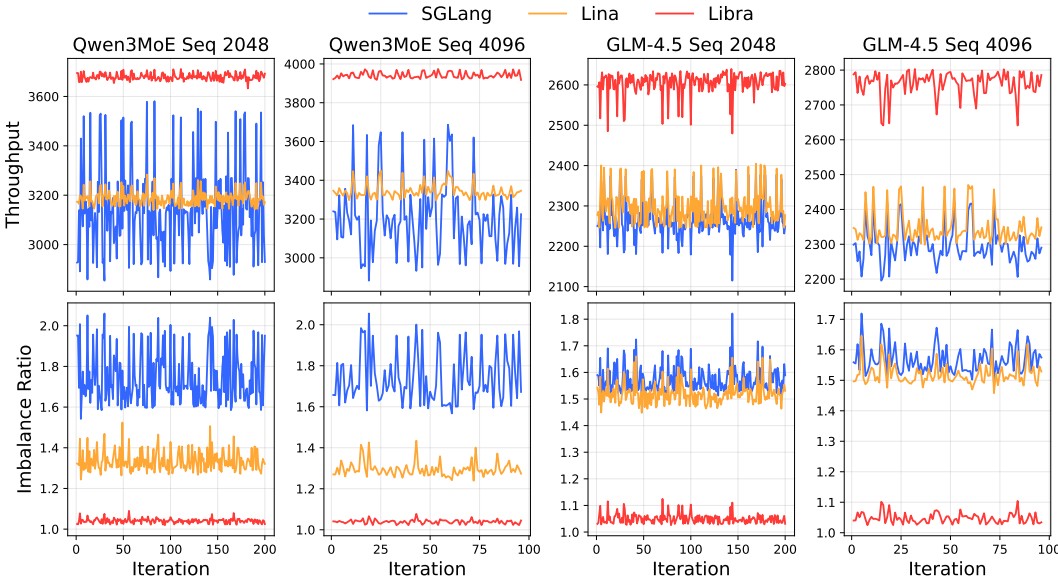

Figure 9: Throughput fluctuation and imbalance ratio under a dynamic workload.

Lina, its expert-selection-path table was constructed using the same dataset as the evaluation and its expert prefetching allowed each GPU to hold 8 additional experts. Similarly, EPLB's expert placement was determined by profiling on the identical dataset. It statically placed 8 identified hot experts—one on each of the 8 GPUs—across every MoE layer. This static replication, however, results in higher memory consumption compared to the dynamic approaches of Lina and Libra because of a large number of layers. To meet the memory budget with Lina, Libra was configured with $N$ set to 8 and $\alpha$ set to 0.5. Despite these favorable, even biased, conditions for the baselines, Libra consistently and significantly outperforms them for both Qwen3MoE and GLM-4.5. These results confirm that Libra's dynamic load balancing effectively resolves the straggler problem, leading to superior computational efficiency and overall system performance.

**Per-Iteration Fluctuation**. Libra also delivers significantly higher and more stable throughput. Figure 9 illustrates this robustness using a mixed-dataset designed to simulate dynamic shifts in expert load imbalance. For this test, Lina's expert-selection-path table was constructed using a workload created by mixing one-eighth of the build split from each of the eight datasets. This comparison centers on dynamic systems like Lina and Libra, excluding EPLB, as its reliance on periodic profiling and static reconfiguration is ill-suited for workloads where imbalance shifts frequently and intensely. While baseline systems suffer from volatile performance that plummets as the imbalance ratio spikes, Libra effectively decouples its performance from the input distribution. By maintaining

| Qwen3MoE | | | | GLM-4.5 | | |
| --- | --- | --- | --- | --- | --- | --- |
| **Dataset** | **Lina** | **Libra** | | **Dataset** | **Lina** | **Libra** |
| BookCorpus | 47.3 | 91.7 | | BookCorpus | 11.7 | 79.6 |
| DeepSeek-Prover-V1 | 45.4 | 86.5 | | DeepSeek-Prover-V1 | 12.7 | 72.9 |
| HellaSwag | 37.5 | 86.6 | | HellaSwag | 11.5 | 76.6 |
| HumanEvalPlus | 44.5 | 87.0 | | HumanEvalPlus | 11.2 | 72.7 |

Table 1: Prediction accuracy.

| Scheme | Dispatch | Predict | Broadcast | D2H metadata transfer | H2D metadata transfer | Token sharding | Repl. planning | Combine | Others | MoE | Total |
| --- | --- | --- | --- | --- | --- | --- | --- | --- | --- | --- | --- |
| Libra | 0.99[a] | 0.03 | 0.24[a] | 0.10[a] | 0.09[a] | 0.57[a] | 0.26[b] | 1.33 | 0.39 | 2.77 (Local) + 4.55 (Remote) | 9.07 |
| Lina | 0.72 | 0.40 | 0 | 0 | 0 | 0.15 | 0.08 | 1.32 | 0.55 | 8.11 | 11.33 |
| SGLang | 0.73 | 0 | 0 | 0 | 0 | 0 | 0 | 1.34 | 0.55 | 10.99 | 13.61 |

[a] Hidden in MoE_local,    [b] Hidden in MoE_remote,

Table 2: Breakdown analysis of MoE layer.

a near-perfect imbalance ratio close to 1.0, Libra provides consistently high throughput, proving its resilience to the dynamic nature of expert load imbalance.

**Prediction Accuracy**. We quantitatively evaluate the accuracy of Libra's predictor for hot expert replication compared against Lina. Table 1 presents a direct comparison between the prediction accuracy of Libra's speculative execution-based approach and Lina's offline-constructed lookup table based method on the Qwen3MoE and GLM-4.5 models. Accuracy is defined as the fraction of correctly predicted experts for each token, where the set of actually activated experts serves as the ground truth. We construct Lina's expert-selection-path table on the build split of mixed dataset, then evaluate on the evaluation split of four datasets. Evaluation setup is detailed in Appendix A.

The results reveal a stark contrast between the two methods. Libra's predictor consistently achieves a high and stable accuracy in the 70-90% range across all datasets, demonstrating the effectiveness of its runtime prediction based on current-layer hidden states. In contrast, Lina's lookup-based predictor shows noticeably lower accuracy across datasets, and highlights the critical generalization limitations of an offline-built lookup table. This effect is more pronounced on GLM-4.5, where Lina successfully identifies fewer than one correct expert between top-8 experts. These findings confirm that Libra's dynamic prediction mechanism is significantly more robust and reliable for handling diverse and unpredictable workloads.

**Breakdown Analysis**. We present the inference latency breakdown of the MoE layer for SGLang, Lina, and Libra in Table 2, evaluated using the Qwen3MoE with sequence length of 1024 and batch size of 32. Both Lina and Libra significantly reduce MoE computation time through load balancing; however, Libra achieves a greater reduction, primarily driven by its superior expert-activation prediction accuracy. Notably, while Lina incurs visible prediction latency, Libra's prediction overhead is negligible. Furthermore, although Libra introduces additional costs associated with its sophisticated execution model—including metadata transfer, broadcasting, and balancing logic—these are effectively hidden. By decoupling MoE computation into $MoE_{local}$ and $MoE_{remote}$, Libra overlaps these auxiliary operations with local computation, preventing them from impacting the critical path.

# 6 CONCLUSION

We introduce Libra, a dynamic load balancing system addressing intensified expert load imbalance in modern Mixture-of-Experts (MoE) models. Libra proposes *Two-Stage Locality-Aware Execution*, an innovative paradigm hiding load balancing overhead by overlapping it with ongoing GPU computations. This is enabled by two synergistic core components: *Locality-Aware Expert Replication*, using speculative execution for highly accurate (70-80%) expert prediction to strategically prefetch the necessary experts for the next layer, and *Adaptive Token Sharding*, computing an optimal as-

signment schedule by accounting for processed local token load. Implemented on SGLang, Libra demonstrates state-of-the-art performance, improving the prefill throughput by up to 19.2% while maintaining an imbalance ratio of nearly 1.0 under dynamic workloads. Libra thus achieves dynamic load balancing with virtually zero-overhead for efficient serving of large-scale MoE models.

ACKNOWLEDGMENTS

This paper was the result of the research project supported by SK hynix Inc. Jae W. Lee is the corresponding author.

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

## A   EXPERIMENTAL SETUP DETAILS

**Models**. For Figure 1, we compare the imbalance ratio across recent MoE families by pairing models with different load-balancing strategies during training. Within the Qwen family, we evaluate Qwen2MoE (Yang et al., 2024) and Qwen3MoE, released on HuggingFace as `Qwen2-57B-A14B` and `Qwen3-235B-A22B`. Qwen2MoE has 57 billion total parameters and 14 billion activated parameters uses 64 experts with top-8 expert selection, and utilizes a micro-batch level auxiliary load-balancing loss during training to ensure load balance. In contrast, Qwen3MoE, with 235 billion total and 22 billion activated parameters, uses 128 experts with top-8 expert selection and forgoes this term in favor of a global-batch load balancing loss, a strategy that maximizes expert specialization while still addressing balance. The DeepSeek family shows a similar contrast: DeepSeek-V2 (236B total, 21B activated) also employs 160 experts with top-6 expert selection and two shared experts, and also uses an auxiliary load-balancing loss, whereas DeepSeek-V3 (671B total, 37B activated) employs 256 expert with top-8 expert selection and a single shared expert, and improves training efficiency by adopting the auxiliary-loss-free load balancing technique. For all other experiments, we evaluate Libra and baselines on Qwen3MoE and GLM-4.5, models that are trained without such fine-grained balancing losses. GLM-4.5 has 355 billion total parameters and 32 billion activated parameters, and uses 160 experts with top-8 expert selection and a single shared expert.

**Datasets**. We evaluate *Libra* and baselines on eight datasets: BookCorpus (Zhu et al., 2015), Codeforces (Penedo et al., 2025), DeepSeek-Prover (Xin et al., 2024), FineWeb (Penedo et al., 2024), GSM8K (Cobbe et al., 2021), HellaSwag (Zellers et al., 2019), HumanEvalPlus (Liu et al., 2023a), and LMSYS-Chat-1M (Zheng et al., 2023). Unless noted otherwise, each dataset contributes a total of 2.0M tokens. The first 1.6M tokens form the build split for EPLB offline profiling and for constructing Lina's prediction table. The next 0.4M tokens form the evaluation split used for testing. Figure 1 is the only exception, which uses a 0.07% subset of the BookCorpus dataset. For Table 1, we build Lina's expert-selection-path table on a mixed workload that uniformly interleaves the build splits of all eight datasets. We then evaluate Lina and Libra separately on each dataset's evaluation split. Figure 8 reports results on BookCorpus, DeepSeek-Prover-V1, HellaSwag, and HumanEvalPlus dataset. Figure 9 uses a shuffled workload constructed from all eight datasets.

**Environments**. All experiments use a single-node system equipped with eight NVIDIA H200-SXM5 GPUs, each with 141GB of HBM3e memory. The server configuration is summarized in Table 3.

| | |
|---|---|
| CPU | $2\times$ Intel Xeon Platinum 8580 (128 cores) |
| GPU | $8\times$ NVIDIA H200-SXM5-141GB |
| System Memory | $32\times$ 64 GB DDR5-5600 (total 2,048 GB) |
| GPU Memory | 141GB HBM3e per GPU |
| GPU Interconnect | Connected with NVSwitch (900GB/s bandwidth) |

Table 3: Server configuration

**Metrics**. Table 1 reports accuracy, defined as the fraction of tokens whose ground-truth experts appear in the predicted top-k set. Figure 8 and Figure 9 report prefill throughput in tokens per second. Throughout these experiments, we adopt Prefill-Decode disaggregation (Patel et al., 2024; Zhong et al., 2024a) setup, and therefore we evaluate prefill only as we target prefill phase. Figure 1 and Figure 9 also report the imbalance ratio, defined as the load of the most heavily utilized GPU divided by the average load across all GPUs.

## B   EXPERT REPLICATION PLANNING

The *Expert Replication Planning* optimally places experts on GPUs for the next MoE layer. This process is crucial for facilitating effective load balancing and extending the $MoE_{local}$ computation window, which in turn hides system overhead. The algorithm operates in two main phases after an initial setup.

First, it duplicates "local hot experts"—those most frequently requested by a GPU's local tokens but residing on other GPUs—onto the source GPU itself. This strategically increases the number

of local tokens that can be processed without inter-GPU communication, creating a sufficient time window for the *Adaptive Token Sharding* to execute in parallel without affecting the critical path.

Second, the algorithm iteratively balances the remaining load by duplicating global hot experts. It identifies the most overloaded GPU and replicates its hottest expert to the least-loaded GPU that has available capacity. This ensures that the overall load is distributed as evenly as possible before the next layer's computation begins. The final output is an optimized expert placement map that serves as the foundation for the rebalancing stage.

---

**Algorithm 1** Expert Replication Planning

---

**Inputs:** Predicted expert IDs for upcoming tokens next_topk_ids, Total number of experts $E$, Total number of GPUs $G$, Max duplicated experts per GPU $N$, Number of local hot experts to duplicate $L$.
**Outputs:** A binary matrix for expert placement on GPUs: $M_{\text{next}} \in \{0, 1\}^{E \times G}$.

1: Calculate ExpertLoad$[g, e]$ (requests for expert $e$ from GPU $g$) based on next_topk_ids.
2: Initialize $M_{\text{next}}$ by assigning each expert to its home GPU.
3: **for** each GPU $g_s$ **do**                ▷ Phase 1: Duplicate Local Hot Experts
4:      Identify top $L$ remote experts most requested by $g_s$.
5:      Duplicate these experts to $g_s$.
6: **end for**
7: Calculate initial GPU loads based on the current mapping in $M_{\text{next}}$.
8: $B \leftarrow$ Target balanced load per GPU.
9: **for** $i = 1$ to $(N - L) \times G$ **do**         ▷ Phase 2: Balance Load via Iterative Duplication
10:      $g_{\text{src}} \leftarrow$ most overloaded GPU where load $> B$.
11:      **if** no such GPU exists **then break**
12:      **end if**
13:      $e \leftarrow$ expert contributing most to $g_{\text{src}}$'s remote load.
14:      $g_{\text{dst}} \leftarrow$ least loaded candidate GPU that can host $e$ (respecting capacity $N$).
15:      **if** $g_{\text{dst}}$ is found and duplicating $e$ keeps Load$[g_{\text{dst}}] \leq B$ **then**
16:          Update $M_{\text{next}}$ by duplicating $e$ to $g_{\text{dst}}$.
17:          Update GPU loads to reflect newly localized computations.
18:      **end if**
19: **end for**
20: **return** $M_{\text{next}}$

---

# C ADAPTIVE TOKEN SHARDING

The *Adaptive Token Sharding* determines an optimal assignment for remote tokens to resolve load imbalance, operating on the expert placement map generated by the *Expert Replication Planning*. Its core strategy is an iterative greedy approach that ensures the final token distribution is as close to perfectly balanced as possible.

The algorithm begins by calculating the total load for each GPU. It then enters a loop that continues as long as any GPU's load exceeds a target threshold. Within the loop, it identifies the most overloaded GPU ($g_s$) and selects its hottest remote expert ($e$)—the one responsible for the largest portion of its remote token load. It then finds the least-loaded GPU ($g_d$) that already hosts a replica of expert $e$ and has sufficient capacity.

A calculated number of tokens for expert $e$ are then transferred from $g_s$ to $g_d$, and the load states of both GPUs are updated. After each transfer, the algorithm restarts its loop to re-evaluate the global system state, ensuring it always addresses the most critical imbalance first. This process repeats until the loads are balanced or no further beneficial transfers can be made.

---

**Algorithm 2** Adaptive Token Sharding

---

**Require:** Expert-to-GPU mapping $M \in \{0,1\}^{E \times G}$, Local GPU loads $L \in \mathbb{Z}^G$, Remote expert loads $R \in \mathbb{Z}^{G \times E}$, Average target load $B \in \mathbb{R}$, Imbalance tolerance $\varepsilon \in (0,1)$
**Ensure:** Rebalanced loads $R, G$, and maximum load $t_{max}$
1: $G[g] \leftarrow L[g] + \sum_e R[g,e]$ for all $g \in G$             ▷ Initialize total GPU loads
2: **while** $\max_g G[g] > (1 + \varepsilon)B$ **do**
3:      moved $\leftarrow$ false
4:      **for** each $g_s \in \{g \mid G[g] > B\}$ in descending order of $G[g]$ **do**
5:          **for** each $e \in \{e \mid R[g_s, e] > 0\}$ in descending order of $R[g_s, e]$ **do**
6:              $\mathcal{C} \leftarrow \{g \neq g_s \mid M[e,g] = 1\}$          ▷ Find candidate destination GPUs
7:              **if** $\mathcal{C}$ is not empty **then**
8:                  $g_d \leftarrow \arg\min_{g \in \mathcal{C}} G[g]$         ▷ Select least-loaded destination
9:                  cap $\leftarrow B - G[g_d]$        ▷ Calculate destination's remaining capacity
10:                  **if** cap $> 0$ **then**
11:                      $l \leftarrow \min(R[g_s, e], \text{cap})$        ▷ Determine amount to move
12:                      $R[g_s, e] \leftarrow R[g_s, e] - l; \quad R[g_d, e] \leftarrow R[g_d, e] + l$
13:                      $G[g_s] \leftarrow G[g_s] - l; \quad G[g_d] \leftarrow G[g_d] + l$    ▷ Perform the token transfer
14:                      moved $\leftarrow$ true
15:                      **break**       ▷ Exit inner loop to re-evaluate the most overloaded GPU
16:                  **end if**
17:              **end if**
18:          **end for**
19:      **end for**
20:      **if** not moved **then**
21:          **break**                  ▷ Converged or stuck, exit outer loop
22:      **end if**
23: **end while**
24: $t_{max} \leftarrow \max_g G[g]$; **return** $R, G, t_{max}$

---

# D    END TO END LATENCY ANALYSIS

Libra prioritizes accelerating the prefill phase to minimize Time-To-First-Token (TTFT), which is crucial for prefill-intensive workloads like Summarization and Retrieval-Augmented Generation (RAG). While the decoding stage often dominates total latency, minimizing TTFT is essential for meeting strict Service-Level Objectives (SLOs) (Du et al., 2025; Xiong et al., 2024; Chen et al., 2025). To demonstrate Libra's effectiveness in this regard, we conducted end-to-end experiments on Qwen3MoE and GLM-4.5 across various configurations, the results of which are detailed in Table 4 through Table 13.

The results demonstrate that Libra significantly reduces TTFT, particularly in long-context scenarios. We note that for short sequences (e.g., 1K tokens), performance may be comparable to or slightly lower than the baseline, as the $MoE_{local}$ window is insufficient to fully mask CPU overhead. However, as input length increases, Libra effectively mitigates the straggler effect, achieving TTFT reductions of up to 27% for Qwen3MoE and 12.7% for GLM-4.5 in large batch and long-context settings.

| Batch | Scheme | Output Length | | | | | | | | |
|---|---|---|---|---|---|---|---|---|---|---|
| | | 1 | 2 | 4 | 8 | 16 | 32 | 64 | 128 | 256 |
| 8 | Libra | 0.38 | 0.50 | 0.73 | 1.22 | 2.17 | 4.08 | 7.87 | 15.62 | 31.05 |
| | Lina | 0.37 | 0.49 | 0.73 | 1.22 | 2.19 | 4.14 | 7.99 | 15.49 | 30.98 |
| | SGLang | 0.36 | 0.48 | 0.72 | 1.19 | 2.14 | 4.04 | 8.00 | 15.60 | 31.07 |
| 16 | Libra | 0.57 | 0.70 | 0.93 | 1.41 | 2.38 | 4.29 | 8.27 | 15.75 | 31.05 |
| | Lina | 0.65 | 0.77 | 1.01 | 1.50 | 2.45 | 4.35 | 8.09 | 16.08 | 31.19 |
| | SGLang | 0.70 | 0.82 | 1.06 | 1.54 | 2.52 | 4.50 | 8.24 | 16.00 | 31.80 |
| 32 | Libra | 1.03 | 1.16 | 1.40 | 1.87 | 2.88 | 4.81 | 8.57 | 16.40 | 32.11 |
| | Lina | 1.22 | 1.35 | 1.58 | 2.06 | 3.03 | 5.02 | 8.96 | 16.54 | 32.51 |
| | SGLang | 1.38 | 1.50 | 1.75 | 2.23 | 3.19 | 5.18 | 8.97 | 16.82 | 32.16 |
| 48 | Libra | 1.51 | 1.63 | 1.86 | 2.33 | 3.27 | 5.18 | 8.96 | 16.16 | 31.07 |
| | Lina | 1.79 | 1.91 | 2.15 | 2.61 | 3.55 | 5.42 | 9.14 | 16.87 | 31.78 |
| | SGLang | 2.04 | 2.16 | 2.39 | 2.87 | 3.81 | 5.66 | 9.43 | 16.99 | 32.05 |
| 64 | Libra | 2.00 | 2.12 | 2.35 | 2.83 | 3.77 | 5.68 | 9.43 | 16.93 | 31.75 |
| | Lina | 2.38 | 2.50 | 2.73 | 3.21 | 4.15 | 5.98 | 9.82 | 16.98 | 32.33 |
| | SGLang | 2.72 | 2.84 | 3.08 | 3.56 | 4.49 | 6.35 | 10.10 | 17.59 | 33.20 |

Table 4: End to end latency for Qwen3MoE, Input length = 1024

| Batch | Scheme | Output Length | | | | | | | | |
|---|---|---|---|---|---|---|---|---|---|---|
| | | 1 | 2 | 4 | 8 | 16 | 32 | 64 | 128 | 256 |
| 8 | Libra | 0.56 | 0.68 | 0.92 | 1.39 | 2.32 | 4.24 | 8.11 | 15.31 | 30.78 |
| | Lina | 0.65 | 0.77 | 1.01 | 1.48 | 2.41 | 4.31 | 8.32 | 15.58 | 30.69 |
| | SGLang | 0.71 | 0.82 | 1.07 | 1.53 | 2.48 | 4.40 | 8.14 | 15.70 | 31.26 |
| 16 | Libra | 1.04 | 1.16 | 1.41 | 1.89 | 2.89 | 4.76 | 8.80 | 16.50 | 31.32 |
| | Lina | 1.24 | 1.36 | 1.59 | 2.10 | 3.03 | 4.99 | 8.89 | 16.50 | 32.01 |
| | SGLang | 1.39 | 1.51 | 1.75 | 2.25 | 3.19 | 5.17 | 8.94 | 16.72 | 32.00 |
| 32 | Libra | 2.02 | 2.13 | 2.37 | 2.86 | 3.84 | 5.76 | 9.58 | 17.10 | 32.53 |
| | Lina | 2.40 | 2.52 | 2.76 | 3.26 | 4.23 | 6.09 | 9.93 | 17.71 | 33.12 |
| | SGLang | 2.75 | 2.87 | 3.12 | 3.59 | 4.56 | 6.47 | 10.42 | 18.07 | 33.41 |
| 48 | Libra | 3.00 | 3.12 | 3.36 | 3.82 | 4.75 | 6.60 | 10.36 | 17.99 | 33.24 |
| | Lina | 3.57 | 3.68 | 3.90 | 4.39 | 5.33 | 7.19 | 11.03 | 18.63 | 33.67 |
| | SGLang | 4.12 | 4.24 | 4.47 | 4.93 | 5.87 | 7.72 | 11.40 | 18.96 | 34.15 |
| 64 | Libra | 4.00 | 4.12 | 4.35 | 4.82 | 5.75 | 7.62 | 11.43 | 18.87 | 33.91 |
| | Lina | 4.76 | 4.87 | 5.11 | 5.58 | 6.50 | 8.39 | 12.12 | 19.39 | 34.79 |
| | SGLang | 5.51 | 5.63 | 5.87 | 6.35 | 7.24 | 9.14 | 12.80 | 20.37 | 34.75 |

Table 5: End to end latency for Qwen3MoE, Input length = 2048

| Batch | Scheme | Output Length | | | | | | | | |
|---|---|---|---|---|---|---|---|---|---|---|
| | | 1 | 2 | 4 | 8 | 16 | 32 | 64 | 128 | 256 |
| 8 | Libra | 1.06 | 1.18 | 1.42 | 1.90 | 2.84 | 4.82 | 8.61 | 16.13 | 31.49 |
| | Lina | 1.27 | 1.38 | 1.62 | 2.10 | 3.06 | 4.98 | 8.73 | 16.35 | 32.00 |
| | SGLang | 1.41 | 1.53 | 1.77 | 2.26 | 3.18 | 5.12 | 9.00 | 16.52 | 32.11 |
| 16 | Libra | 2.06 | 2.18 | 2.42 | 2.90 | 3.89 | 5.76 | 9.62 | 17.46 | 32.31 |
| | Lina | 2.44 | 2.56 | 2.80 | 3.29 | 4.24 | 6.23 | 10.05 | 17.81 | 33.10 |
| | SGLang | 2.80 | 2.92 | 3.16 | 3.64 | 4.60 | 6.58 | 10.25 | 18.13 | 33.68 |
| 32 | Libra | 4.08 | 4.20 | 4.43 | 4.92 | 5.85 | 7.78 | 11.62 | 19.21 | 34.70 |
| | Lina | 4.82 | 4.93 | 5.18 | 5.66 | 6.61 | 8.54 | 12.33 | 20.17 | 34.93 |
| | SGLang | 5.60 | 5.72 | 5.96 | 6.44 | 7.44 | 9.38 | 13.08 | 20.83 | 35.80 |
| 48 | Libra | 6.12 | 6.22 | 6.48 | 6.92 | 7.91 | 9.75 | 13.57 | 21.03 | 36.04 |
| | Lina | 7.24 | 7.35 | 7.60 | 8.08 | 9.05 | 10.88 | 14.71 | 22.11 | 37.39 |
| | SGLang | 8.41 | 8.54 | 8.79 | 9.21 | 10.21 | 12.03 | 15.97 | 23.42 | 38.35 |
| 64 | Libra | 8.19 | 8.30 | 8.54 | 9.01 | 9.93 | 11.88 | 15.65 | 23.12 | 37.73 |
| | Lina | OOM | OOM | OOM | OOM | OOM | OOM | OOM | OOM | OOM |
| | SGLang | 11.25 | 11.36 | 11.60 | 12.08 | 12.98 | 14.79 | 18.59 | 26.07 | 40.78 |

Table 6: End to end latency for Qwen3MoE, Input length = 4096

| Batch | Scheme | Output Length | | | | | | | | |
| | | 1 | 2 | 4 | 8 | 16 | 32 | 64 | 128 | 256 |
|---|---|---|---|---|---|---|---|---|---|---|
| 8 | Libra | 1.59 | 1.71 | 1.94 | 2.41 | 3.38 | 5.28 | 9.15 | 16.60 | 31.36 |
| | Lina | 1.88 | 2.00 | 2.24 | 2.71 | 3.69 | 5.58 | 9.35 | 17.13 | 31.43 |
| | SGLang | 2.13 | 2.25 | 2.49 | 2.96 | 3.89 | 5.82 | 9.58 | 17.17 | 31.90 |
| 16 | Libra | 3.13 | 3.26 | 3.49 | 3.98 | 4.91 | 6.89 | 10.83 | 18.49 | 34.38 |
| | Lina | 3.69 | 3.81 | 4.05 | 4.54 | 5.48 | 7.50 | 11.26 | 19.10 | 34.68 |
| | SGLang | 4.25 | 4.38 | 4.61 | 5.10 | 6.03 | 8.01 | 11.95 | 19.61 | 35.50 |
| 32 | Libra | 6.24 | 6.37 | 6.59 | 7.10 | 8.05 | 9.98 | 13.75 | 21.57 | 36.71 |
| | Lina | 7.36 | 7.48 | 7.72 | 8.20 | 9.17 | 11.06 | 15.10 | 22.40 | 37.67 |
| | SGLang | 8.54 | 8.67 | 8.89 | 9.40 | 10.35 | 12.28 | 16.05 | 23.88 | 39.01 |
| 48 | Libra | 10.13 | 10.24 | 10.48 | 10.93 | 11.86 | 13.70 | 17.52 | 24.42 | 39.65 |
| | Lina | OOM | OOM | OOM | OOM | OOM | OOM | OOM | OOM | OOM |
| | SGLang | 12.82 | 12.94 | 13.15 | 13.62 | 14.55 | 16.35 | 20.18 | 27.55 | 42.22 |
| 64 | Libra | 12.87 | 12.98 | 13.21 | 13.67 | 14.63 | 16.52 | 20.28 | 27.64 | 42.63 |
| | Lina | OOM | OOM | OOM | OOM | OOM | OOM | OOM | OOM | OOM |
| | SGLang | 17.15 | 17.26 | 17.49 | 17.95 | 18.86 | 20.74 | 24.51 | 31.88 | 46.94 |

Table 7: End to end latency for Qwen3MoE, Input length = 6144

| Batch | Scheme | Output Length | | | | | | | | |
| | | 1 | 2 | 4 | 8 | 16 | 32 | 64 | 128 | 256 |
|---|---|---|---|---|---|---|---|---|---|---|
| 8 | Libra | 2.14 | 2.26 | 2.50 | 2.98 | 3.90 | 5.80 | 9.60 | 16.97 | 32.07 |
| | Lina | 2.53 | 2.65 | 2.90 | 3.39 | 4.35 | 6.29 | 10.07 | 17.88 | 33.25 |
| | SGLang | 2.88 | 3.00 | 3.23 | 3.70 | 4.62 | 6.56 | 10.28 | 17.94 | 32.83 |
| 16 | Libra | 4.25 | 4.37 | 4.61 | 5.09 | 6.05 | 8.03 | 12.05 | 19.61 | 35.39 |
| | Lina | 5.00 | 5.12 | 5.36 | 5.86 | 6.82 | 8.72 | 12.70 | 20.33 | 35.40 |
| | SGLang | 5.77 | 5.90 | 6.14 | 6.64 | 7.61 | 9.51 | 13.33 | 21.20 | 36.70 |
| 32 | Libra | 8.53 | 8.65 | 8.89 | 9.37 | 10.36 | 12.26 | 16.22 | 23.98 | 39.99 |
| | Lina | OOM | OOM | OOM | OOM | OOM | OOM | OOM | OOM | OOM |
| | SGLang | 11.59 | 11.72 | 11.94 | 12.43 | 13.44 | 15.39 | 19.28 | 27.18 | 42.86 |
| 48 | Libra | 13.13 | 13.24 | 13.48 | 13.94 | 14.86 | 16.69 | 20.41 | 27.65 | 42.48 |
| | Lina | OOM | OOM | OOM | OOM | OOM | OOM | OOM | OOM | OOM |
| | SGLang | 17.40 | 17.52 | 17.74 | 18.20 | 19.15 | 20.99 | 24.68 | 32.15 | 47.12 |
| 64 | Libra | OOM | OOM | OOM | OOM | OOM | OOM | OOM | OOM | OOM |
| | Lina | OOM | OOM | OOM | OOM | OOM | OOM | OOM | OOM | OOM |
| | SGLang | 23.21 | 23.33 | 23.58 | 24.06 | 25.07 | 27.01 | 30.77 | 38.70 | 54.52 |

Table 8: End to end latency for Qwen3MoE, Input length = 8192

| Batch | Scheme | Output Length | | | | | | | | |
| | | 1 | 2 | 4 | 8 | 16 | 32 | 64 | 128 | 256 |
|---|---|---|---|---|---|---|---|---|---|---|
| 8 | Libra | 0.48 | 0.63 | 0.92 | 1.51 | 2.70 | 5.12 | 9.58 | 19.35 | 37.85 |
| | Lina | 0.65 | 0.80 | 1.10 | 1.66 | 2.84 | 5.25 | 9.87 | 19.20 | 38.12 |
| | SGLang | 0.44 | 0.59 | 0.87 | 1.45 | 2.63 | 5.02 | 9.71 | 19.16 | 38.22 |
| 16 | Libra | 0.79 | 0.93 | 1.23 | 1.83 | 3.02 | 5.38 | 10.18 | 19.78 | 39.51 |
| | Lina | 0.91 | 1.06 | 1.37 | 1.96 | 3.16 | 5.66 | 10.23 | 19.89 | 39.41 |
| | SGLang | 0.84 | 0.99 | 1.29 | 1.90 | 3.07 | 5.47 | 10.23 | 19.89 | 39.26 |
| 32 | Libra | 1.48 | 1.63 | 1.94 | 2.54 | 3.73 | 6.19 | 10.75 | 20.98 | 40.08 |
| | Lina | 1.73 | 1.88 | 2.19 | 2.79 | 3.99 | 6.41 | 11.07 | 20.77 | 39.92 |
| | SGLang | 1.65 | 1.81 | 2.10 | 2.70 | 3.91 | 6.35 | 11.24 | 20.80 | 40.61 |
| 48 | Libra | 2.17 | 2.32 | 2.62 | 3.22 | 4.40 | 6.75 | 11.47 | 21.30 | 39.95 |
| | Lina | 2.55 | 2.70 | 2.99 | 3.61 | 4.79 | 7.17 | 11.85 | 21.34 | 40.85 |
| | SGLang | 2.45 | 2.60 | 2.89 | 3.49 | 4.69 | 7.13 | 12.02 | 21.33 | 40.87 |
| 64 | Libra | 2.88 | 3.03 | 3.33 | 3.91 | 5.05 | 7.47 | 12.10 | 21.82 | 41.01 |
| | Lina | 3.39 | 3.54 | 3.84 | 4.43 | 5.60 | 8.02 | 12.65 | 22.32 | 40.81 |
| | SGLang | 3.28 | 3.43 | 3.73 | 4.32 | 5.47 | 7.78 | 12.47 | 21.97 | 41.15 |

Table 9: End to end latency for GLM-4.5, Input length = 1024

| Batch | Scheme | Output Length | | | | | | | | |
|---|---|---|---|---|---|---|---|---|---|---|
| | | 1 | 2 | 4 | 8 | 16 | 32 | 64 | 128 | 256 |
| 8 | Libra | 0.80 | 0.95 | 1.25 | 1.86 | 3.01 | 5.37 | 10.18 | 19.55 | 38.71 |
| | Lina | 0.92 | 1.07 | 1.36 | 1.96 | 3.15 | 5.52 | 10.12 | 20.25 | 39.68 |
| | SGLang | 0.84 | 0.99 | 1.29 | 1.87 | 3.05 | 5.49 | 10.25 | 19.80 | 38.85 |
| 16 | Libra | 1.50 | 1.66 | 1.95 | 2.54 | 3.75 | 6.14 | 10.92 | 20.65 | 39.45 |
| | Lina | 1.74 | 1.89 | 2.19 | 2.78 | 3.97 | 6.36 | 11.32 | 20.40 | 40.12 |
| | SGLang | 1.67 | 1.82 | 2.11 | 2.72 | 3.91 | 6.30 | 10.99 | 20.66 | 39.23 |
| 32 | Libra | 2.92 | 3.07 | 3.36 | 3.98 | 5.22 | 7.68 | 12.36 | 22.29 | 41.89 |
| | Lina | 3.41 | 3.56 | 3.86 | 4.47 | 5.66 | 8.12 | 12.81 | 22.61 | 41.13 |
| | SGLang | 3.31 | 3.46 | 3.76 | 4.36 | 5.54 | 8.03 | 12.66 | 22.47 | 41.41 |
| 48 | Libra | OOM | OOM | OOM | OOM | OOM | OOM | OOM | OOM | OOM |
| | Lina | OOM | OOM | OOM | OOM | OOM | OOM | OOM | OOM | OOM |
| | SGLang | 4.95 | 5.10 | 5.38 | 5.98 | 7.13 | 9.55 | 14.37 | 23.63 | 42.89 |
| 64 | Libra | OOM | OOM | OOM | OOM | OOM | OOM | OOM | OOM | OOM |
| | Lina | OOM | OOM | OOM | OOM | OOM | OOM | OOM | OOM | OOM |
| | SGLang | OOM | OOM | OOM | OOM | OOM | OOM | OOM | OOM | OOM |

Table 10: End to end latency for GLM-4.5, Input length = 2048

| Batch | Scheme | Output Length | | | | | | | | |
|---|---|---|---|---|---|---|---|---|---|---|
| | | 1 | 2 | 4 | 8 | 16 | 32 | 64 | 128 | 256 |
| 8 | Libra | 1.54 | 1.69 | 1.99 | 2.57 | 3.76 | 6.12 | 10.70 | 20.18 | 38.81 |
| | Lina | 1.77 | 1.92 | 2.22 | 2.80 | 3.98 | 6.31 | 11.02 | 20.53 | 39.38 |
| | SGLang | 1.70 | 1.84 | 2.14 | 2.71 | 3.92 | 6.33 | 11.10 | 20.49 | 40.07 |
| 16 | Libra | 2.99 | 3.14 | 3.44 | 4.04 | 5.23 | 7.71 | 12.50 | 21.87 | 41.61 |
| | Lina | 3.47 | 3.62 | 3.93 | 4.53 | 5.76 | 8.13 | 12.87 | 22.25 | 41.88 |
| | SGLang | 3.37 | 3.52 | 3.82 | 4.40 | 5.59 | 8.15 | 12.88 | 22.49 | 41.10 |
| 32 | Libra | OOM | OOM | OOM | OOM | OOM | OOM | OOM | OOM | OOM |
| | Lina | OOM | OOM | OOM | OOM | OOM | OOM | OOM | OOM | OOM |
| | SGLang | OOM | OOM | OOM | OOM | OOM | OOM | OOM | OOM | OOM |
| 48 | Libra | OOM | OOM | OOM | OOM | OOM | OOM | OOM | OOM | OOM |
| | Lina | OOM | OOM | OOM | OOM | OOM | OOM | OOM | OOM | OOM |
| | SGLang | OOM | OOM | OOM | OOM | OOM | OOM | OOM | OOM | OOM |
| 64 | Libra | OOM | OOM | OOM | OOM | OOM | OOM | OOM | OOM | OOM |
| | Lina | OOM | OOM | OOM | OOM | OOM | OOM | OOM | OOM | OOM |
| | SGLang | OOM | OOM | OOM | OOM | OOM | OOM | OOM | OOM | OOM |

Table 11: End to end latency for GLM-4.5, Input length = 4096

| Batch | Scheme | Output Length | | | | | | | | |
|---|---|---|---|---|---|---|---|---|---|---|
| | | 1 | 2 | 4 | 8 | 16 | 32 | 64 | 128 | 256 |
| 8 | Libra | 2.31 | 2.46 | 2.75 | 3.37 | 4.50 | 6.90 | 11.63 | 21.13 | 40.36 |
| | Lina | 2.65 | 2.80 | 3.10 | 3.69 | 4.89 | 7.26 | 11.96 | 21.48 | 40.80 |
| | SGLang | 2.56 | 2.71 | 3.00 | 3.62 | 4.75 | 7.15 | 11.88 | 21.38 | 40.61 |
| 16 | Libra | OOM | OOM | OOM | OOM | OOM | OOM | OOM | OOM | OOM |
| | Lina | OOM | OOM | OOM | OOM | OOM | OOM | OOM | OOM | OOM |
| | SGLang | OOM | OOM | OOM | OOM | OOM | OOM | OOM | OOM | OOM |
| 32 | Libra | OOM | OOM | OOM | OOM | OOM | OOM | OOM | OOM | OOM |
| | Lina | OOM | OOM | OOM | OOM | OOM | OOM | OOM | OOM | OOM |
| | SGLang | OOM | OOM | OOM | OOM | OOM | OOM | OOM | OOM | OOM |
| 48 | Libra | OOM | OOM | OOM | OOM | OOM | OOM | OOM | OOM | OOM |
| | Lina | OOM | OOM | OOM | OOM | OOM | OOM | OOM | OOM | OOM |
| | SGLang | OOM | OOM | OOM | OOM | OOM | OOM | OOM | OOM | OOM |
| 64 | Libra | OOM | OOM | OOM | OOM | OOM | OOM | OOM | OOM | OOM |
| | Lina | OOM | OOM | OOM | OOM | OOM | OOM | OOM | OOM | OOM |
| | SGLang | OOM | OOM | OOM | OOM | OOM | OOM | OOM | OOM | OOM |

Table 12: End to end latency for GLM-4.5, Input length = 6144

| Batch | Scheme | Output Length | | | | | | | | |
|---|---|---|---|---|---|---|---|---|---|---|
| | | 1 | 2 | 4 | 8 | 16 | 32 | 64 | 128 | 256 |
| 8 | Libra | 3.05 | 3.19 | 3.49 | 4.08 | 5.25 | 7.67 | 12.32 | 22.05 | 40.76 |
| | Lina | 3.59 | 3.74 | 4.03 | 4.63 | 5.82 | 8.23 | 13.06 | 22.44 | 41.51 |
| | SGLang | 3.49 | 3.64 | 3.92 | 4.53 | 5.70 | 7.98 | 12.76 | 22.41 | 41.73 |
| 16 | Libra | OOM | OOM | OOM | OOM | OOM | OOM | OOM | OOM | OOM |
| | Lina | OOM | OOM | OOM | OOM | OOM | OOM | OOM | OOM | OOM |
| | SGLang | OOM | OOM | OOM | OOM | OOM | OOM | OOM | OOM | OOM |
| 32 | Libra | OOM | OOM | OOM | OOM | OOM | OOM | OOM | OOM | OOM |
| | Lina | OOM | OOM | OOM | OOM | OOM | OOM | OOM | OOM | OOM |
| | SGLang | OOM | OOM | OOM | OOM | OOM | OOM | OOM | OOM | OOM |
| 48 | Libra | OOM | OOM | OOM | OOM | OOM | OOM | OOM | OOM | OOM |
| | Lina | OOM | OOM | OOM | OOM | OOM | OOM | OOM | OOM | OOM |
| | SGLang | OOM | OOM | OOM | OOM | OOM | OOM | OOM | OOM | OOM |
| 64 | Libra | OOM | OOM | OOM | OOM | OOM | OOM | OOM | OOM | OOM |
| | Lina | OOM | OOM | OOM | OOM | OOM | OOM | OOM | OOM | OOM |
| | SGLang | OOM | OOM | OOM | OOM | OOM | OOM | OOM | OOM | OOM |

Table 13: End to end latency for GLM-4.5, Input length = 8192

