# OpenReview forum: "Libra: Effective yet Efficient Load Balancing for Large-scale MoE Inference"
_ICLR.cc/2026/Conference — ICLR 2026 Poster_

### Official Review · Reviewer_SZWt · 2025-10-29

**Soundness:** 3
**Presentation:** 3
**Contribution:** 3
**Rating:** 8
**Confidence:** 5

**Summary:**

This paper addresses the critical challenge of expert load imbalance in distributed Mixture-of-Experts (MoE) model inference. The authors propose Libra, a system design that combines accurate prediction of expert activations with sophisticated load balancing algorithms while effectively hiding overhead through careful execution flow reconstruction. The system achieves up to 19.2% throughput improvement on state-of-the-art MoE models.

**Strengths:**

* This work is well-motivated by demonstrating that recent SoTA MoE models (DeepSeek-V3, Qwen3MoE, GLM-4.5) exhibit intensified load imbalance compared to their predecessors

* The combination of algorithm observation (slow evolution of hidden states) and system design is novel and insightful.

* The experiments cover multiple models, diverse datasets, and demonstrate both static and dynamic workload performance.

* The paper is well-written with effective visualizations that make the technical approach accessible.

**Weaknesses:**

* Despite the overall "slow evolution of hidden states", how does the evolution vary acorss different MoE layers? For example, the early MoE layers could result in more significant hidden state dynamics, which might make the proposed method less effective.

* I would love to see how Libra scales to multi-node deployments

**Questions:**

see above

---

> ### Author Response · Authors · 2025-11-21
>
> We thank the reviewer for the helpful suggestions. We have responded to each point raised in your review below.
>
> > Despite the overall "slow evolution of hidden states", how does the evolution vary across different MoE layers? For example, the early MoE layers could result in more significant hidden state dynamics, which might make the proposed method less effective.
> >
>
> As the reviewer hypothesized, the dynamics of hidden states do vary depending on the depth of the layer. To quantify this, we measured the **cosine similarity between the hidden states of adjacent MoE layers** alongside the **prediction accuracy of our Lookahead predictor** across different layers. We note that the starting layers for measurement differ primarily due to model architectures. **Qwen3MoE** consists entirely of MoE layers, whereas **GLM-4.5** begins with 3 dense FFN layers before the first MoE layer. Consequently, the table presents data starting from **Layer 1** (the 2nd layer) for Qwen3MoE and **Layer 4** (the 5th layer) for GLM-4.5.
>
> As shown in the Table 1, our analysis confirms a strong correlation between hidden state dynamics and prediction accuracy. In the early layers, we observe lower cosine similarity due to the rapid evolution of feature representations, which leads to relatively lower prediction accuracy. However, this "transient phase" accounts for only a small fraction (approximately 3-4%) of the total model depth. As the hidden states rapidly stabilize—indicated by cosine similarity rising to 0.9–0.95—our predictor achieves high accuracy for the vast majority of the layers. Consequently, Libra operates with full effectiveness during the dominant "stable phase," ensuring that the temporary inefficiency in the initial layers has a negligible impact on end-to-end throughput.
>
> **Table 1: Cosine similarity of hidden states between adjacent MoE layers and prediction accuracy.**
>
> Qwen3-235B-A22B (BookCorpus)
>
> | Layer | 1 | 2 | 3 | 4 | 5 | 6 | 7 | 8 | 9 | 10 | 20 | 30 | 40 | 50 | 60 | 70 | 80 | 90 |
> | --- | --- | --- | --- | --- | --- | --- | --- | --- | --- | --- | --- | --- | --- | --- | --- | --- | --- | --- |
> | Prediction accuracy (%) | 65.61 | 77.08 | 76.68 | 83.39 | 82.35 | 86.52 | 87.99 | 90.22 | 89.28 | 88.84 | 92.54 | 89.34 | 93.11 | 93.23 | 92.83 | 94.53 | 93.31 | 93.54 |
> | Cosine similarity | 0.71 | 0.80 | 0.81 | 0.89 | 0.89 | 0.91 | 0.92 | 0.94 | 0.93 | 0.95 | 0.97 | 0.98 | 0.96 | 0.96 | 0.96 | 0.94 | 0.92 | 0.96 |
>
> Qwen3-235B-A22B (DeepSeek-Prover-V1)
>
> | Layer | 1 | 2 | 3 | 4 | 5 | 6 | 7 | 8 | 9 | 10 | 20 | 30 | 40 | 50 | 60 | 70 | 80 | 90 |
> | --- | --- | --- | --- | --- | --- | --- | --- | --- | --- | --- | --- | --- | --- | --- | --- | --- | --- | --- |
> | Prediction accuracy (%) | 64.44 | 61.50 | 59.74 | 75.71 | 78.47 | 78.10 | 79.70 | 83.69 | 84.46 | 83.83 | 88.22 | 86.19 | 89.53 | 89.12 | 86.36 | 87.91 | 83.70 | 87.66 |
> | Cosine similarity | 0.72 | 0.80 | 0.83 | 0.90 | 0.90 | 0.92 | 0.92 | 0.94 | 0.93 | 0.95 | 0.97 | 0.97 | 0.96 | 0.95 | 0.94 | 0.92 | 0.90 | 0.96 |
>
> Qwen3-235B-A22B (HellaSwag)
>
> | Layer | 1 | 2 | 3 | 4 | 5 | 6 | 7 | 8 | 9 | 10 | 20 | 30 | 40 | 50 | 60 | 70 | 80 | 90 |
> | --- | --- | --- | --- | --- | --- | --- | --- | --- | --- | --- | --- | --- | --- | --- | --- | --- | --- | --- |
> | Prediction accuracy (%) | 67.25 | 58.85 | 54.47 | 66.26 | 70.53 | 78.27 | 76.02 | 82.59 | 82.96 | 85.35 | 89.50 | 86.38 | 90.20 | 88.62 | 86.90 | 89.68 | 86.70 | 89.25 |
> | Cosine similarity | 0.65 | 0.68 | 0.77 | 0.85 | 0.86 | 0.92 | 0.91 | 0.93 | 0.94 | 0.95 | 0.98 | 0.97 | 0.97 | 0.96 | 0.94 | 0.93 | 0.90 | 0.95 |
>
> Qwen3-235B-A22B (HumanEvalPlus)
>
> | Layer | 1 | 2 | 3 | 4 | 5 | 6 | 7 | 8 | 9 | 10 | 20 | 30 | 40 | 50 | 60 | 70 | 80 | 90 |
> | --- | --- | --- | --- | --- | --- | --- | --- | --- | --- | --- | --- | --- | --- | --- | --- | --- | --- | --- |
> | Prediction accuracy (%) | 67.96 | 61.82 | 58.85 | 68.34 | 76.69 | 77.68 | 80.74 | 80.34 | 85.55 | 83.56 | 88.94 | 85.64 | 88.51 | 90.46 | 87.56 | 89.22 | 86.71 | 87.79 |
> | Cosine similarity | 0.76 | 0.76 | 0.76 | 0.89 | 0.90 | 0.90 | 0.90 | 0.90 | 0.90 | 0.93 | 0.98 | 0.98 | 0.97 | 0.96 | 0.95 | 0.92 | 0.90 | 0.95 |

---

> > ### Author Response · Authors · 2025-11-21
> >
> > (cont'd)
> >
> > GLM-4.5 (BookCorpus)
> >
> > | Layer | 4 | 5 | 6 | 7 | 8 | 9 | 10 | 20 | 30 | 40 | 50 | 60 | 70 | 80 | 90 |
> > | --- | --- | --- | --- | --- | --- | --- | --- | --- | --- | --- | --- | --- | --- | --- | --- |
> > | Prediction accuracy (%) | 51.75 | 60.15 | 55.06 | 66.06 | 65.53 | 61.32 | 61.14 | 79.16 | 78.97 | 80.09 | 83.62 | 86.84 | 86.96 | 85.24 | 85.70 |
> > | Cosine similarity | 0.87 | 0.94 | 0.94 | 0.95 | 0.93 | 0.92 | 0.93 | 0.95 | 0.91 | 0.92 | 0.96 | 0.98 | 0.98 | 0.97 | 0.97 |
> >
> > GLM-4.5 (DeepSeek-Prover-V1)
> >
> > | Layer | 4 | 5 | 6 | 7 | 8 | 9 | 10 | 20 | 30 | 40 | 50 | 60 | 70 | 80 | 90 |
> > | --- | --- | --- | --- | --- | --- | --- | --- | --- | --- | --- | --- | --- | --- | --- | --- |
> > | Prediction accuracy (%) | 47.96 | 57.16 | 56.00 | 61.99 | 60.59 | 52.04 | 59.99 | 70.64 | 70.30 | 68.95 | 76.99 | 82.59 | 83.43 | 78.39 | 78.19 |
> > | Cosine similarity | 0.87 | 0.94 | 0.91 | 0.93 | 0.90 | 0.93 | 0.92 | 0.94 | 0.90 | 0.90 | 0.95 | 0.98 | 0.99 | 0.96 | 0.97 |
> >
> > GLM-4.5 (HellaSwag)
> >
> > | Layer | 4 | 5 | 6 | 7 | 8 | 9 | 10 | 20 | 30 | 40 | 50 | 60 | 70 | 80 | 90 |
> > | --- | --- | --- | --- | --- | --- | --- | --- | --- | --- | --- | --- | --- | --- | --- | --- |
> > | Prediction accuracy (%) | 50.15 | 47.75 | 55.57 | 64.21 | 60.89 | 55.85 | 64.05 | 76.29 | 77.45 | 75.45 | 83.84 | 86.72 | 84.99 | 80.09 | 81.25 |
> > | Cosine similarity | 0.90 | 0.94 | 0.92 | 0.93 | 0.91 | 0.93 | 0.91 | 0.95 | 0.91 | 0.92 | 0.97 | 0.98 | 0.98 | 0.95 | 0.96 |
> >
> > GLM-4.5 (HumanEvalPlus)
> >
> > | Layer | 4 | 5 | 6 | 7 | 8 | 9 | 10 | 20 | 30 | 40 | 50 | 60 | 70 | 80 | 90 |
> > | --- | --- | --- | --- | --- | --- | --- | --- | --- | --- | --- | --- | --- | --- | --- | --- |
> > | Prediction accuracy (%) | 53.80 | 59.06 | 58.38 | 66.00 | 58.89 | 51.66 | 58.50 | 73.29 | 68.66 | 68.10 | 72.70 | 81.89 | 83.86 | 78.31 | 74.92 |
> > | Cosine similarity | 0.88 | 0.95 | 0.92 | 0.94 | 0.91 | 0.95 | 0.93 | 0.95 | 0.89 | 0.91 | 0.95 | 0.98 | 0.99 | 0.95 | 0.96 |
> >
> > > I would love to see how Libra scales to multi-node deployments
> > >
> >
> > We believe the core principles of Libra—specifically the mechanisms for token sharding and replication planning—can be extended to multi-node settings with appropriate adaptations.
> >
> > Specifically, the following extensions would be necessary:
> >
> > 1. **Bandwidth-Aware Token Sharding:** Currently, our *Token Sharding* mechanism (Section 3.3) employs an iterative greedy strategy to optimize GPU load balancing. In a multi-node setting, where inter-node interconnects (e.g., InfiniBand) have significantly lower bandwidth than intra-node links (e.g., NVLink), we need to incorporate communication bandwidth into the cost model. This will allow the system to prioritize routing tokens to experts within the same node, explicitly treating inter-node bandwidth as a scarce resource.
> > 2. **Hierarchical Expert Replication Planning:** Our *Expert Replication Planning* (Section 3.2) currently optimizes locality by duplicating local hot experts to each GPU. We need to extend this to a hierarchical model. By carefully balancing the ratio of **intra-node replication** versus **inter-node replication**, we can maximize the effective utilization of the cluster's hierarchical network bandwidth, minimizing expensive cross-node transfers.
> >
> > A full multi-node implementation is beyond the scope of what can be added during the rebuttal period, but the extensions we outline above provide a direct and feasible path to supporting multi-node deployments in future work.

---

> ### Comment · Reviewer_SZWt · 2025-11-23
>
> Interesting results! I appreciate the authors' response and maintain my rating of 8.

---

> > ### Author Response · Authors · 2025-11-24
> >
> > We thank the reviewer for the positive feedback and the confirmation of the rating. We are glad that the breakdown of hidden state evolution across layers addressed your question effectively. Thank you for your insightful review and support.

---

### Official Review · Reviewer_knvi · 2025-11-01

**Soundness:** 3
**Presentation:** 2
**Contribution:** 3
**Rating:** 4
**Confidence:** 4

**Summary:**

This paper addresses the issue of expert load imbalance during MoE model inference by implementing runtime expert placement based on load prediction.
It achieves the effective integration and systematic orchestration of the proposed methods, which are built upon existing methods and offer enhancements.
Experimental results demonstrate significant end-to-end inference speedup on the evaluated MoE models.

**Strengths:**

1. This manuscript is well-structured, making it easy to follow.
2. It effectively integrates the proposed designs in the system implementation, resulting in significant efficiency improvements.

**Weaknesses:**

1. Lack of detailed breakdown analysis regarding the duration of operations and system overhead, including GPU-CPU data transfer overhead as well as the execution times of operations on both the GPU and CPU.
2. Lack of analysis on various model and system configurations.
3. The work demonstrates relatively modest innovation, primarily through the integration of established optimization strategies, such as overlapping using local expert computation, load prediction, and expert placement.

**Questions:**

Given that various model configurations (e.g., hidden size, number of experts, top-k activation) and system configurations (e.g., CPU/GPU computational and communication capabilities) influence the duration of specific operations and, consequently, affect the overlap of asynchronous processes, it is essential to provide a detailed breakdown analysis. Such an analysis can elucidate the extent of the optimization space and identify the scenarios in which these optimizations are most beneficial.

---

> ### Author Response · Authors · 2025-11-21
>
> Thank you for your constructive feedback. As requested, we have included the detailed breakdown analysis in Section 5. We respond to your specific comments in the section below.
>
> > 1. Lack of detailed breakdown analysis regarding the duration of operations and system overhead, including GPU-CPU data transfer overhead as well as the execution times of operations on both the GPU and CPU.
> >
>
> > 2. Lack of analysis on various model and system configurations.
> >
>
> > Question: Given that various model configurations (e.g., hidden size, number of experts, top-k activation) and system configurations (e.g., CPU/GPU computational and communication capabilities) influence the duration of specific operations and, consequently, affect the overlap of asynchronous processes, it is essential to provide a detailed breakdown analysis. Such an analysis can elucidate the extent of the optimization space and identify the scenarios in which these optimizations are most beneficial.
> >
>
> ### **Latency Breakdown**
>
> We present the inference latency breakdown of the MoE layer for SGLang, Lina, and Libra using the Qwen3-235B-A22B model in Table 1.
>
> The components are defined as follows:
>
> - **Dispatch / Combine / MoE:** Standard MoE operations. For Libra, we explicitly separate the MoE computation into $MoE_{local}$ and $MoE_{remote}$ to demonstrate how the overheads are hidden.
> - **Predict:** Time taken for the predictor (next layer gating function for Libra or table lookup for Lina).
> - **Broadcast:** Time to broadcast top-k expert IDs to all GPUs.
> - **H2D / D2H Metadata Transfer:** Latency for transferring metadata required for/generated by the CPU-based balancing logics.
> - **Token Sharding / Replication Planning:** Execution time of the balancing algorithms.
> - **Others:** Cumulative latency of minor operations such as RMSNorm.
>
> Both Lina and Libra significantly reduce MoE time through load balancing, with Libra achieving larger reductions primarily due to its higher expert-activation prediction accuracy. Libra’s prediction overhead is negligible, as opposed to Lina’s, highlighting its lightweight nature. Although Libra introduces additional costs such as metadata transfer, broadcast, and extended token sharding and replication planning due to its more sophisticated algorithms, these costs are effectively mitigated through overlap with MoE time.
>
> **Table 1. MoE Layer Latency Breakdown (ms) (Sequence Length = 1024, Batch Size = 32)**
>
> | Scheme | Dispatch | Predict | Broadcast | D2H metadata transfer | H2D metadata transfer | Token sharding | Replication planning | Combine | Others | MoE | Total |
> | --- | --- | --- | --- | --- | --- | --- | --- | --- | --- | --- | --- |
> | Libra | 0.99 (Hidden in MoE_local) |  0.03  | 0.36 (Hidden in MoE_local) | 0.10 (Hidden in MoE_local) | 0.09 (Hidden in MoE_local) |  0.57 (Hidden in MoE_local) | 0.26 (Hidden in MoE_remote) | 1.33 | 0.39 | 2.77 (MoE_local) + 4.55 (MoE_remote) | 9.07 |
> | Lina | 0.72  | 0.40  | 0 | 0 | 0 | 0.15 | 0.08 | 1.32 | 0.55 | 8.11 | 11.33 |
> | Baseline(SGLang) | 0.73  | 0 | 0 | 0 | 0 | 0 | 0 | 1.34 | 0.54 | 10.99 | 13.61 |
>
> ### **Impact of Model Configurations**
>
> We analyzed the sensitivity of Libra's CPU-side operations—specifically **Token Sharding** and **Replication Planning**—against key model hyperparameters to identify how different model configurations affect the duration of optimization processes.
>
> It is crucial to clarify that the computational overhead of these balancing algorithms is independent of the model's *hidden size*. Instead, the latency is primarily determined by the *total number of experts (*$E$*)* and the *number of activated experts (*$K$*)*. The dependency on $K$ manifests only in simple per-token iterations (e.g., scanning the top-$K$ expert IDs for each token), resulting in linear $O(K)$ work. In contrast, $E$ directly influences the sorting and balancing phases, which operate on per-expert load statistics and exhibit quadratic complexity $O(E^2)$. Therefore, when scaling model configurations, $E$ becomes the dominant factor affecting CPU latency, whereas $K$ contributes a comparatively minor linear overhead.
>
> To empirically verify the impact of $E$, we compared the latency of both operations on the evaluated models, which share the same Top-k ($K=8$) but differ significantly in the number of experts: **Qwen3MoE** ($E=128$) and **GLM-4.5** ($E=160$).
>
> As shown in **Table 2**, although GLM-4.5 possesses 25% more experts than Qwen3MoE (160 vs. 128), the increase in latency for Token Sharding and Replication Planning is negligible. This is because, in practice, the effective cost of the $O(E^2)$ routines is significantly mitigated by early termination in balancing loops that bound the number of iterations. This demonstrates that Libra's planning overhead scales moderately with the number of experts.

---

> ### Author Response · Authors · 2025-11-21
>
> (cont'd)
>
> **Table 2. CPU overheads based on model configurations (ms)**
>
> Token Sharding, Batch Size = 8
>
> |  | 1024 | 2048 | 4096 | 6144 | 8192 |
> | --- | --- | --- | --- | --- | --- |
> | Qwen3MoE | 0.181 | 0.322 | 0.571 | 0.885 | 1.106 |
> | GLM-4.5 | 0.22 | 0.338 | 0.68 | 0.958 | 1.303 |
>
> Replication Planning, Batch Size = 8
>
> |  | 1024 | 2048 | 4096 | 6144 | 8192 |
> | --- | --- | --- | --- | --- | --- |
> | Qwen3MoE | 0.131 | 0.197 | 0.261 | 0.365 | 0.464 |
> | GLM-4.5 | 0.142 | 0.203 | 0.314 | 0.355 | 0.472 |
>
> ### **Impact of System Configurations**
>
> Following the model configuration analysis, we addressed the reviewer's concern regarding **System Configurations**, specifically the balance between CPU and GPU capabilities. Since the effectiveness of Libra relies on hiding the CPU-based overhead within the GPU computation window ($MoE_{local}$), the ratio of CPU-to-GPU performance is a critical factor.
>
> Crucially, **Token Sharding** is the primary CPU operation requiring hiding within the $MoE_{local}$ window. Note that replication planning is excluded here as it effectively overlaps with the subsequent $MoE_{remote}$ phase.
>
> To rigorously evaluate the robustness of this hiding mechanism without the constraints of physical hardware swapping, we adopted a methodology of **CPU frequency scaling** on our experimental node equipped with an Intel(R) Xeon(R) Gold 6258R CPU @ 2.70GHz. By throttling the CPU frequency while keeping the GPU capacity fixed (H200), we effectively simulate scenarios with varying CPU-to-GPU capability ratios.
>
> In our measurements, we tracked the latency of **Token Sharding** against the duration of the $MoE_{local}$ phase under different CPU frequencies. Additionally, we accounted for auxiliary overheads (denoted as 'Others'), which primarily consist of Dispatch and Metadata transfer, to provide a comprehensive breakdown of the hidden operations.
>
> As demonstrated in Table 3, the total overhead (Token Sharding + Others) consistently remains within the $MoE_{local}$ computation window, even under significant CPU throttling. This confirms that the substantial time buffer provided by the heavy matrix multiplication effectively hides the minimal latency of our optimized algorithms, ensuring robust, zero-overhead load balancing across diverse system configurations.
>
> **Table 3. CPU overhead based on system configurations (ms)**
>
> | Sequence length | Token Sharding w/ 100% frequency | Token Sharding w/ 75% frequency | Token Sharding w/ 50% frequency | Token Sharding w/ 25% frequency | Others | MoE_local |
> | --- | --- | --- | --- | --- | --- | --- |
> | 2048 | 0.222 | 0.219 | 0.384 | 0.627 | 0.79 | 1.378 |
> | 4096 | 0.361 | 0.445 | 0.714 | 1.034 | 1.42 | 2.77 |
> | 8192 | 0.775 | 0.879 | 1.015 | 1.795 | 2.47 | 5.436 |

---

### Official Review · Reviewer_HMhF · 2025-11-01

**Soundness:** 3
**Presentation:** 1
**Contribution:** 2
**Rating:** 2
**Confidence:** 5

**Summary:**

The paper addresses the problem of load imbalance in mixture of expert large language model (LLM) inference. We propose a system named Libra that achieves efficient load balancing. Libra builds upon the expert replication policy used in Lina, incorporating an improved mechanism for predicting hot experts. Furthermore, we redesign the execution flow to utilize the CPU effectively, minimizing overhead associated with token sharding.

**Strengths:**

1. The proposed method addresses several drawbacks of Lina. It introduces an improved hot expert predictor and a more efficient expert replication strategy.
2. Additionally, Libra utilizes the CPU to reduce overhead associated with token sharding operations.

**Weaknesses:**

1. Novelty concern: Utilizing prefetching to identify expert activation in the next layer is not a novel approach. It has been previously used for accelerating Mixture-of-Expert (MoE) inference as described in [1]. In addition, [1] also introduces the use of the CPU to prevent lightweight computations from becoming a bottleneck in GPU processing.

2. Token sharding introduces additional communication overhead, which is not addressed by the authors. This issue could become significant as batch sizes increase.

3. The experiments include only two MoE models, which is limited. Newer MoE models, such as Qwen3, Moonlight-A3B, gpt-oss-120b, and gpt-oss-20b, should be considered.

4. In throughput experiments, the models only execute the prefill stage, whereas the decoding stage is the main time-consuming component. The experiments should focus more on the decoding stage.

5, The experiments are conducted with a setup of 8 batch size and 4k sequence length. Larger batch sizes, as well as shorter and longer sequence lengths, should be tested.

6. The study lacks end-to-end latency experiments that include both prefilling and decoding stages with various inference configurations, such as different batch sizes, input sequence lengths, decoding sequence lengths, and datasets.

7. Lastly, the paper does not provide an analysis of the MoE layer inference time.

[1] Pre-gated MoE: An Algorithm-System Co-Design for Fast and Scalable Mixture-of-Expert Inference

**Questions:**

Please refer to the weakness.

---

> ### Author Response · Authors · 2025-11-21
>
> We appreciate your constructive comments. As requested, we have included the additional end-to-end experiments in Appendix D. We respond to individual points from your review below.
>
> > 1. Novelty concern: Utilizing prefetching to identify expert activation in the next layer is not a novel approach. It has been previously used for accelerating Mixture-of-Expert (MoE) inference as described in [1]. In addition, [1] also introduces the use of the CPU to prevent lightweight computations from becoming a bottleneck in GPU processing.
> >
>
> While some components of Libra (e.g., predictor design, CPU usage) may appear similar to [1] Pre-gated MoE at high-level, as the two systems tackle different research goals (MoE load balancing vs. overcoming GPU memory capacity limitations), their underlying mechanisms differ substantially.
>
> - **Predictor Design:**
>     - **Pre-gated MoE** requires **modifying the model architecture** by adding a pre-gate component and **fine-tuning** the model for downstream tasks.
>     - **Libra** is a **training-free, system-level solution** applicable to pre-trained models without modification. It leverages the existing gating function of the next layer for prediction.
> - **CPU Usage:**
>     - **Pre-gated MoE:** Uses the CPU primarily as a **parameter store and data provider**. It does not offload computational logic; the pre-gate computation itself still occurs on the GPU.
>     - **Libra:** Uses the CPU to execute **load balancing logic (replication planning & token sharding)**, effectively offloading these control-plane overheads to achieve zero-overhead balancing on the GPU.
>
> In addition, while predictor design and CPU usage are part of our contributions, Libra also introduces several additional key contributions, as outlined below.
>
> 1. **Two-Stage Locality-Aware Execution:** We decouple the MoE layer into $MoE_{local}$ (no communication needed) and $MoE_{remote}$. We utilize the execution time of $MoE_{local}$ to perform replication planning and token sharding on the CPU, hiding these overheads completely.
> 2. **Locality-Aware Expert Replication:** We prioritize replicating "local hot experts" first to extend the $MoE_{local}$ window, securing sufficient time to hide the CPU balancing overhead.
> 3. **Adaptive Token Sharding:** We distribute only the remaining tokens not processed in $MoE_{local}$, considering the workload already processed during $MoE_{local}$, ensuring global load balance.
>
> > 2. Token sharding introduces additional communication overhead, which is not addressed by the authors. This issue could become significant as batch sizes increase.
> >
>
> We clarify two points regarding the overhead:
>
> 1. **Communication Volume:** Libra **does not increase** the inter-GPU communication volume. Token sharding merely determines the *destination GPUs* of tokens that are *already* designated for remote experts. It optimizes *where* these tokens go to balance the load, but the total number of tokens transferred over the network remains unchanged compared to standard routing.
> 2. **Computation Overhead:** The computation for token sharding (deciding assignments) is performed on the CPU. As detailed in our response to the **breakdown analysis**, this computation overlaps entirely with the $MoE_{local}$ computation on the GPU.
> Furthermore, as the batch size increases, the $MoE_{local}$ computation time (our "hiding window") also increases. This provides *more* slack for the CPU to perform balancing logics, making Libra even more robust at larger batch sizes.
>
> > 3. The experiments include only two MoE models, which is limited. Newer MoE models, such as Qwen3, Moonlight-A3B, gpt-oss-120b, and gpt-oss-20b, should be considered.
> >
>
> The two models we evaluate, Qwen3-235B-A22B and GLM-4.5, already represent state-of-the-art large-scale MoE architectures. For Qwen3-235B-A22B, we use the latest publicly available version (released July 21, 2025), and GLM-4.5 was released shortly thereafter (July 28, 2025). Given that the paper was submitted in September, these models reflect the most up-to-date MoE designs available at the time of submission.
> Regarding the models you mentioned: Qwen3 is already included in our evaluation. Other models such as Moonlight-A3B or gpt-oss-20b / gpt-oss-120b are significantly smaller in scale. Our study focuses on very large MoE models (200B+), where inter-GPU load balancing becomes challenging.

---

> > ### Author Response · Authors · 2025-11-21
> >
> > (cont'd)
> > > 4. In throughput experiments, the models only execute the prefill stage, whereas the decoding stage is the main time-consuming component. The experiments should focus more on the decoding stage.
> > >
> >
> > > 5. The experiments are conducted with a setup of 8 batch size and 4k sequence length. Larger batch sizes, as well as shorter and longer sequence lengths, should be tested.
> > >
> >
> > > 6. The study lacks end-to-end latency experiments that include both prefilling and decoding stages with various inference configurations, such as different batch sizes, input sequence lengths, decoding sequence lengths, and datasets.
> > >
> >
> > Libra's primary goal is to accelerate the **Prefill phase** to strictly minimize **Time-To-First-Token (TTFT)**. Although the decoding stage often dominates total latency, accelerating the prefill stage is critical for **prefill-intensive workload** such as **Summarization** and **Retrieval-Augmented Generation(RAG)**. As highlighted in [2] PrefillOnly, [3] LayerKV, [4] IMPRESS, minimizing TTFT is essential for meeting strict Service-Level Objectives (SLOs) and ensuring user responsiveness in these scenarios.
> >
> > Although our focus is TTFT, we conducted the requested end-to-end experiments for Qwen3-235B-A22B and GLM-4.5 across various configurations. We have included the results in **Appendix D of the revised paper.** As shown in the result, Libra significantly improves TTFT. We frankly note that for very short input sequences (e.g., 1K tokens), the $MoE_{local}$ window may be too brief to fully hide the CPU overhead, resulting in performance comparable to or slightly lower than the baseline. However, Libra's benefits become substantial as input length increases. **Specifically, in long-context and large batch size scenarios, Libra achieves a TTFT reduction of up to 27.5% for Qwen3MoE and 12.7% for GLM-4.5 compared to the baseline.**
> >
> > > 7. Lastly, the paper does not provide an analysis of the MoE layer inference time.
> > >
> >
> > We present the inference latency breakdown of the MoE layer for SGLang, Lina, and Libra using the Qwen3-235B-A22B model in Table 1.
> >
> > The components are defined as follows:
> >
> > - **Dispatch / Combine / MoE:** Standard MoE operations. For Libra, we explicitly separate the MoE computation into $MoE_{local}$ and $MoE_{remote}$ to demonstrate how the overheads are hidden.
> > - **Predict:** Time taken for the predictor (next layer gating function for Libra or table lookup for Lina).
> > - **Broadcast:** Time to broadcast top-k expert IDs to all GPUs.
> > - **H2D / D2H Metadata Transfer:** Latency for transferring metadata required for/generated by the CPU-based balancing logics.
> > - **Token Sharding / Replication Planning:** Execution time of the balancing algorithms.
> > - **Others:** Cumulative latency of minor operations such as RMSNorm.
> >
> > Both Lina and Libra significantly reduce MoE time through load balancing, with Libra achieving larger reductions primarily due to its higher expert-activation prediction accuracy. Libra’s prediction overhead is negligible, as opposed to Lina’s, highlighting its lightweight nature. Although Libra introduces additional costs such as metadata transfer, broadcast, and extended token sharding and replication planning due to its more sophisticated algorithms, these costs are effectively mitigated through overlap with MoE time.
> >
> > **Table 1. MoE Layer Latency Breakdown (ms) (Sequence Length = 1024, Batch Size = 32)**
> >
> > | Scheme | Dispatch | Predict | Broadcast | D2H metadata transfer | H2D metadata transfer | Token sharding | Replication planning | Combine | Others | MoE | Total |
> > | --- | --- | --- | --- | --- | --- | --- | --- | --- | --- | --- | --- |
> > | Libra | 0.99 (Hidden in MoE_local) |  0.03  | 0.24 (Hidden in MoE_local) | 0.10 (Hidden in MoE_local) | 0.09 (Hidden in MoE_local) |  0.57 (Hidden in MoE_local) | 0.26 (Hidden in MoE_remote) | 1.33 | 0.39 | 2.77 (MoE_local) + 4.55 (MoE_remote) | 9.07 |
> > | Lina | 0.72  | 0.40  | 0 | 0 | 0 | 0.15 | 0.08 | 1.32 | 0.55 | 8.11 | 11.33 |
> > | Baseline(SGLang) | 0.73  | 0 | 0 | 0 | 0 | 0 | 0 | 1.34 | 0.54 | 10.99 | 13.60 |
> >
> > [1] Hwang, Ranggi, et al. "Pre-gated MoE: An Algorithm-System Co-Design for Fast and Scalable Mixture-of-Expert Inference." *Proceedings of the ACM/IEEE 51st Annual International Symposium on Computer Architecture.* 2024.
> >
> > [2] Du, Kuntai, et al. “PrefillOnly: An Inference Engine for Prefill-only Workloads in Large Language Model Applications.” *Proceedings of the ACM SIGOPS 31st Symposium on Operating Systems Principles.* 2025.
> >
> > [3] Xiong, Yi, et al. “LayerKV: Optimizing Large Language Model Serving with Layer-wise KV Cache Management.” *arXiv preprint arXiv:2410.00428* (2024).
> >
> > [4] Chen, Weijian, et al. “IMPRESS: An Importance-Informed Multi-Tier Prefix KV Storage System for Large Language Model Inference.” *23rd USENIX Conference on File and Storage Technologies*. 2025.

---

### Official Review · Reviewer_BzVM · 2025-11-01

**Soundness:** 3
**Presentation:** 2
**Contribution:** 3
**Rating:** 6
**Confidence:** 5

**Summary:**

This paper introduces Libra, a system for efficient and effective load balancing during large-scale Mixture-of-Experts (MoE) inference. It addresses the expert load imbalance problem, where certain experts become bottlenecks due to uneven token distribution across GPUs.

Key Contributions:
1. Two-Stage Locality-Aware Execution: Splits MoE computation into local and remote phases, enabling overlap of load balancing overhead with computation.
2. Accurate Expert Prediction: Uses speculative execution based on hidden states to predict expert activations for the next layer, achieving 70–90% accuracy.
3. Efficient Expert Replication & Token Sharding: Introduces locality-aware expert prefetching and CPU-based token rebalancing, improving throughput by up to 19.2% over strong baselines like Lina.
4. Strong Experimental Results: Demonstrates consistent performance gains across multiple datasets and MoE models (Qwen3MoE, GLM-4.5), especially under dynamic workloads.

Overall, Libra achieves near-optimal load balance with minimal overhead, making it a practical and scalable solution for MoE inference.

**Strengths:**

1. The problem in this paper is clearly defined and strongly motivated. It focuses on the expert load imbalance issue during the inference phase of large-scale MoE models, explicitly pointing out that it introduces the “straggler effect”, severely degrading system throughput.
Through experiments (Figure 1), it demonstrates that modern MoE models (e.g., Qwen3MoE, DeepSeek-V3) abandon load-balancing during training in pursuit of expert specialization, exacerbating load imbalance at inference time.
2. The system is elegantly designed and highly innovative. It proposes Two-Stage Locality-Aware Execution, dividing MoE computation into local (MoElocal) and remote (MoEremote) phases, cleverly hiding the overhead introduced by load balancing. It introduces a hidden-state-based speculative expert-activation prediction mechanism, achieving significantly higher accuracy (70–90% vs. 20–30%) compared to static routing tables used in methods like Lina. It designs two modules—Hierarchical Expert Prefetcher and Adaptive Token Rebalancer—to tackle expert replication and token redistribution, respectively, with clear structure and strong synergy.
3. Comprehensive experiments and significant improvements. Experiments in this paper are implemented on SGLang, the LLM serving engine with state-of-the-art efficiency, and evaluations are conducted on two large-scale MoE models (Qwen3MoE and GLM-4.5) and multiple datasets. Libra achieves up to 19.2% higher throughput than the strongest baseline (Lina). Under dynamic-load scenarios, Libra demonstrates high stability, whereas methods like Lina exhibit noticeable performance fluctuations (Figure 9). Detailed experimental configurations and reproduction information—including models, datasets, hardware environments, and evaluation metrics—are provided, enhancing reproducibility.

**Weaknesses:**

1. The memory consumption is not examined in this paper. While the expert replication mechanism is introduced in this paper for balanced workload on distributed devices, this would lead to higher memory consumption, since individual experts can also contribute to a certain degree of memory consumption. A detailed breakdown analysis of memory consumption on each rank would be helpful to examine the performance more systematically.
2. The prediction accuracy in this paper should be better evaluated. In this submission, the predictors are separately trained on different datasets with specialized task domains. While in real LLM serving applications, the task domain may vary across a broad range for different users at different times. Therefore, a more rational approach would be training the predictor on a large-scale dataset with a diverse data distribution and evaluating the prediction accuracy on massive downstream tasks to better evaluate the out-of-distribution (OOD) performance of Libra.

**Questions:**

1. A more detailed and precise explanation of the model configuration should be provided for each model used in this paper. For example, there is no model named Qwen2MoE in HuggingFace.
2. The definition of the imbalance ratio should be placed in the main text rather than the Appendix for better clarification.

---

> ### Author Response · Authors · 2025-11-21
>
> We appreciate the reviewer for the valuable feedback. We have addressed the individual points from your review below.
>
> > 1. The memory consumption is not examined in this paper. While the expert replication mechanism is introduced in this paper for balanced workload on distributed devices, this would lead to higher memory consumption, since individual experts can also contribute to a certain degree of memory consumption. A detailed breakdown analysis of memory consumption on each rank would be helpful to examine the performance more systematically.
> >
>
> In the Table 1, we present a detailed breakdown of memory usage on an 8-GPU cluster. Since the additional memory consumption is governed by the hyperparameter $N$, which specifies the maximum number of replicated experts per GPU, we report results while varying $N$. Note that we use $N=8$ as the default setting in our main experiments to balance performance and memory overhead. Overall, the memory increase remains modest: approximately **1–2%** on the H200 cluster and **2–4%** on the H100 cluster.
>
> Table 1. Memory overhead breakdown
>
> | Qwen3MoE | N=4 | N=6 | N=8 | N=10 | N=12 |
> | --- | --- | --- | --- | --- | --- |
> | H100 (80GB) | 1.76 | 1.93 | 2.11 | 2.29 | 2.46 |
> | H200 (141GB) | 1.00 | 1.10 | 1.20 | 1.30 | 1.40 |
>
> | GLM-4.5 | N=4 | N=6 | N=8 | N=10 | N=12 |
> | --- | --- | --- | --- | --- | --- |
> | H100 (80GB) | 2.64 | 2.86 | 3.08 | 3.30 | 3.52 |
> | H200 (141GB) | 1.50 | 1.62 | 1.75 | 1.87 | 1.99 |
>
> > 2. The prediction accuracy in this paper should be better evaluated. In this submission, the predictors are separately trained on different datasets with specialized task domains. While in real LLM serving applications, the task domain may vary across a broad range for different users at different times. Therefore, a more rational approach would be training the predictor on a large-scale dataset with a diverse data distribution and evaluating the prediction accuracy on massive downstream tasks to better evaluate the out-of-distribution (OOD) performance of Libra.
> >
>
> First, we would like to clarify that Libra’s predictor does not require any training. Libra directly uses the LLM’s own gating function from the next layer, applied to the current layer’s hidden states, as its prediction mechanism.
>
> The setting you mentioned applies to Lina’s predictor, which is one of our comparison baselines. Because we aim for a strong baseline to highlight Libra’s superiority, we performed task-specific predictor training for Lina. Note that even with such a strong baseline, Libra still achieves better performance.
>
> > A more detailed and precise explanation of the model configuration should be provided for each model used in this paper. For example, there is no model named Qwen2MoE in HuggingFace.
> >
>
> We apologize for the confusion caused by the shorthand names. We have updated **Appendix A** to explicitly map these shorthands to their official HuggingFace model IDs and detailed configurations. For example:
>
> - **Qwen2MoE** $\rightarrow$ **Qwen2-57B-A14B** (Total 64 experts, Top-8 selection).
> - **Qwen3MoE** $\rightarrow$ **Qwen3-235B-A22B** (Total 128 experts, Top-8 selection).
> - **GLM-4.5** (Total 160 experts, Top-8 selection + 1 shared expert).
> - **DeepSeek-V2** (Total 160 experts, Top-6 selection + 2 shared experts).
> - **DeepSeek-V3** (Total 256 experts, Top-8 selection + 1 shared expert).
>
> > The definition of the imbalance ratio should be placed in the main text rather than the Appendix for better clarification.
> >
>
> The definition is currently provided in Section 2.1 as well as Section 5.1 ("defined as the maximum load on any single GPU divided by the average load across all GPUs").

---

### Meta-Review · Area_Chair_B5eX · 2025-12-16

**Summary:**

This paper proposes Libra, an inference system aimed at balancing the "effectiveness" of load balancing and the "overhead of the mechanism itself" to address expert load imbalance (stragglers) in large-scale Mixture-of-Experts (MoE) inference. The core designs are: (i) Two-Stage Locality-Aware Execution, which splits MoE computation into MoE_local / MoE_remote, thereby removing load balancing operations (expert replication planning / token sharding) from the critical path and hiding them in parallel with MoE_local; and (ii) a lookahead prediction mechanism for next-layer gating based on the observation that hidden state changes between layers are gradual, allowing for high-precision estimation of next-layer expert activation to perform locality-aware replication and sharding. Implemented on SGLang using 8×H200 GPUs with Qwen3-235B-A22B / GLM-4.5, the paper reports a maximum throughput improvement of 19.2%. In their response, the authors reinforced the paper with memory overhead breakdowns, MoE layer latency breakdowns (including CPU/GPU and transfer), additional end-to-end evaluations (focusing on TTFT), layer-wise hidden state dynamics and prediction accuracy, and CPU throttling resilience. However, validation in a multi-node setting remains an item for future work.

**Reviewer Concerns:**

- **Novelty/Differentiation (vs. Pre-gated MoE, etc., and existing work like Lina):** HMhF expressed strong concern that "next-layer activation lookahead + CPU utilization is similar to existing work." The authors explained that while Pre-gated MoE requires model modification and fine-tuning, Libra is a training-free inference-time system solution focused on execution flow reconfiguration (Two-Stage) and locality-aware planning. However, as this is a system-integration contribution, evaluations of its originality may vary.
- **Insufficient Evaluation (prefill-heavy/end-to-end, breakdown, sensitivity analysis):** HMhF, knvi, and BzVM requested decode/end-to-end metrics, detailed breakdowns, and sensitivity to settings. The authors presented additional end-to-end experiments (focusing on TTFT improvement), MoE layer latency breakdowns (dispatch/prediction/transfer/planning/Combine, etc.), and sensitivity analyses for model/system configurations (seq length, number of experts, CPU frequency throttling) in the Appendix and main text, resolving most issues. It was noted, however, that short inputs might lack sufficient overlap windows, leaving the possibility that improvements in overall decoding depend on specific settings.
- **Memory Overhead:** BzVM was concerned about the additional memory required for replication. The authors showed a breakdown by varying the cap on replicated experts, reporting approximately 1–2% overhead on H200 (and 2–4% on H100).
- **Layer-wise Dynamics / Generality of Prediction Accuracy:* SZWt pointed out the impact of large hidden state changes in early layers. The authors presented layer-wise cosine similarity and prediction accuracy, explaining that while accuracy drops in early layers, these represent only a small portion of all layers. Regarding BzVM's concern about "changing task distributions," the authors clarified that Libra's prediction does not require additional training.
- **Scale (multi-node):** SZWt requested multi-node experiments. The authors limited their response to describing a design extension plan, leaving experimental verification unresolved.

**Reviewer Scores:**

- **Reviewer BzVM (Initial 6 / conf 5):** Major concerns (memory breakdown, setting clarification, positioning of prediction) were largely resolved with specific data and explanations, so **6→6 maintained (High confidence)**.
- **Reviewer HMhF (Initial 2 / conf 5):** Fundamental concerns regarding novelty (déjà vu regarding lookahead/CPU utilization) partially remain, but the additional end-to-end evaluation and latency breakdown/sensitivity analysis were substantial. Predicted **2→4 (Medium confidence)** (probability of rising to 6+ is low).
- **Reviewer knvi (Initial 4 / conf 4):** The requested detailed breakdown (including CPU/GPU/transfer) and sensitivity analysis for model/system configurations were presented, significantly alleviating technical concerns. While the view that innovation is "relatively modest" for an integrated system remains, it is judged to be within the acceptance range as a practical system contribution for this track. Predicted **4→6 (Medium confidence)**.
- **Reviewer SZWt (Initial 8 / conf 5):** Explicitly stated "**maintain my rating of 8**" in the discussion, so **8→8 (Confirmed)** is adopted.

Predicted Average: (6 + 4 + 6 + 8) / 4 = **6.0**.

---

### Decision · Program_Chairs · 2026-01-26

Accept (Poster)